# Aggregation of Dependent Expert Distributions in Multimodal Variational Autoencoders

**Rogelio A Mancisidor** [1]  **Robert Jenssen** [2 3 4]  **Shujian Yu** [2 5]  **Michael Kampffmeyer** [2 4]

## Abstract

Multimodal learning with variational autoencoders (VAEs) requires estimating joint distributions to evaluate the evidence lower bound (ELBO). Current methods, the product and mixture of experts, aggregate single-modality distributions assuming independence for simplicity, which is an overoptimistic assumption. This research introduces a novel methodology for aggregating single-modality distributions by exploiting the principle of *consensus of dependent experts* (CoDE), which circumvents the aforementioned assumption. Utilizing the CoDE method, we propose a novel ELBO that approximates the joint likelihood of the multimodal data by learning the contribution of each subset of modalities. The resulting CoDE-VAE model demonstrates better performance in terms of balancing the trade-off between generative coherence and generative quality, as well as generating more precise log-likelihood estimations. CoDE-VAE further minimizes the generative quality gap as the number of modalities increases. In certain cases, it reaches a generative quality similar to that of unimodal VAEs, which is a desirable property that is lacking in most current methods. Finally, the classification accuracy achieved by CoDE-VAE is comparable to that of state-of-the-art multimodal VAE models. CoDE-VAE is available at: https://github.com/rogelioamancisidor/codevae.

---

[1]Department of Data Science, BI Norwegian Business School, Oslo, Norway [2]Department of Physics and Technology, UiT The Arctic University of Norway, Tromsø, Norway [3]Department of Computer Science, University of Copenhagen, Copenhagen, Denmark [4]Dept. BAMJO, Norwegian Computing Center, Oslo, Norway [5]Department of Computer Science, Vrije Universiteit Amsterdam, Amsterdam, Netherlands. Correspondence to: Rogelio A Mancisidor <rogelio.a.mancisidor@bi.no>.

*Proceedings of the $42^{nd}$ International Conference on Machine Learning*, Vancouver, Canada. PMLR 267, 2025. Copyright 2025 by the author(s).

## 1. Introduction

Motivated by the observation that learned representations from multimodal data tend to be more generalizable (Wu & Goodman, 2018; 2019), variational autoencoders (VAEs) (Kingma & Welling, 2014; Rezende et al., 2014) have been used to learn representations from multiple data modalities as they are capable of simultaneously generating new observations and inferring joint representations. However, during model inference, it is not guaranteed that we will have access to all modalities, as multimodal data is expensive to obtain (Wu & Goodman, 2018; 2019; Sutter et al., 2021). Therefore, multimodal VAEs should be able to generate representations even when some modalities are missing. This fact has led to a line of research on scalable multimodal generative models with VAEs (Wu & Goodman, 2018; Shi et al., 2019; Sutter et al., 2021; Hwang et al., 2021; Palumbo et al., 2023) in which the number of single-modality distributions, called *experts*, is linear to the number of modalities $M$.

The product of experts (PoE) (Hinton, 2002) and mixture of experts (MoE) are currently the predominant methods to estimate joint variational posterior distributions in multimodal VAEs. However, both assume independence between the experts for simplicity. This is an overoptimistic assumption, as the data modalities are simply different sources of information on the same object. Besides, both have flaws in the estimation of the joint posterior. PoE can produce posterior distributions with lower density if any of the experts also has low density, and if the precision of the experts is miscalibrated, the results can be biased (Shi et al., 2019). The weights in MoE, on the other hand, can be seen as probabilities that the distribution is correct. However, the notion of a "correct model" is not reasonable in this context (Winkler, 1981), as the objective is to aggregate estimates from expert distributions. Furthermore, MoE is inefficient in high-dimensional spaces, as the joint posterior cannot be sharper than any of the experts (Hinton, 2002). In addition to the aforementioned limitations in estimating the joint variational posterior distributions, the optimization of multimodal VAEs requires a valid evidence lower bound (ELBO). In the past, this has been obtained by summing the ELBOs corresponding to each of the $k$-th subsets

$\mathbb{X}_k \in \mathcal{P}(\mathbb{X})$ (Sutter et al., 2021), where $\mathbb{X}$ is the multimodal data, $\mathcal{P}(\mathbb{X})$ is its powerset, and where each $k$-th ELBO contributes equally to optimization. We argue that not all subsets provide the same information and, therefore, should not contribute equally to optimization.

Methods for aggregation of expert distributions, such as PoE or MoE, are called consensus of experts (Winkler, 1981). Although the information provided by expert distributions can exhibit dependence, limited attention has been paid to it. Therefore, this research advances the method introduced in (Winkler, 1981) by extending it to multivariate data and applying it to the approximation of joint variational distributions in multimodal VAEs, which is a nontrivial task. Our proposed consensus of dependent experts (CoDE) method expresses the dependence between experts through the experts' error of estimation and estimates the joint variational distributions with a principled Bayesian method, which circumvents the notion of a correct model and the approximation of vaguer posterior distributions than any of the experts. Furthermore, when the expert distributions are miscalibrated, or uncertain, the joint distribution estimated by CoDE leans towards expert distributions that are more certain. Based on CoDE, we introduce a novel multimodal VAE model (CoDE-VAE) that not only takes into account the dependence between expert distributions when estimating joint posterior variational distributions, but also learns the contribution of each $k$-th ELBO term in optimizing the overall ELBO. An important distinction in estimating joint posterior distributions and optimizing the CoDE-VAE model is that it does not rely on sub-sampling[1] (Daunhawer et al., 2022), which has been shown to harm the approximation of the joint distribution of the data and yet is used by some multimodal VAEs.

The empirical analysis in this research shows that, when dependence between experts is considered, CoDE-VAE exhibits better performance in terms of balancing the tradeoff between generative coherence and generative quality, as well as generating more precise log-likelihood estimations. Furthermore, CoDE-VAE minimizes the generative quality gap as the number of modalities increases, achieving unconditional FID scores similar to unimodal VAEs, which is a desirable property that is lacking in most current models. Finally, CoDE-VAE achieves a classification accuracy that is comparable to that of current state-of-the-art multimodal VAEs. The contributions of this research are: (1) the development of a novel consensus of experts method that takes into account the stochastic dependence between experts; (2) the use of this novel consensus of experts method to derive a new ELBO that learns the contribution of each subset of modalities to optimization; (3) to show that the

generative coherence and generative quality of generated modalities, as well as likelihood approximations in multimodal VAEs, are higher when the dependence between experts is considered and the contribution of ELBO terms to optimization is learned.

## 2. Related Work

The initial research on multimodal learning with VAEs focused on bimodal data, e.g., (Suzuki et al., 2017; Vedantam et al., 2018) where the consensus distribution is approximated by concatenating the two modalities or (Wang et al., 2017) where the authors simply approximate the two expert distributions, assuming that one modality is available during training and test time. Since then, there have been different lines of research to improve performance in multimodal VAEs. For example, the information on the class labels has been used to improve the performance in discriminative tasks (Tsai et al., 2019; Mancisidor et al., 2024) and to disentangle sources of variation (Ilse et al., 2020); information-theoretic approaches have modeled the total correlation (Hwang et al., 2021), common information (Kleinman et al., 2023), and mutual information (Mancisidor et al., 2024) in the objective function; latent distributions have been aligned by minimizing Wasserstein distances (Theodoridis et al., 2020) and using adversarial networks (Chen & Zhu, 2021); (Joy et al., 2022) use mutual supervision to combine information from modalities; hierarchical representation levels of latent representations are used to improve generative properties (Vasco et al., 2022); modeling modality-specific and shared latent distributions has been widely used (Hsu & Glass, 2018; Sutter et al., 2020; Lee & Pavlovic, 2020; Palumbo et al., 2023); diffusion decoders and clustering techniques have been coupled with multimodal VAEs (Palumbo et al., 2024).

One of the fundamental problems in multimodal VAEs is how to approximate joint variational posterior distributions. Joint representations must capture the abstract compositional structure of the underlying object (Wu & Goodman, 2019), and the approximation method must be flexible to scale to a large number of modalities. The predominant methods for this task are PoE (Wu & Goodman, 2018; Sutter et al., 2021; Hwang et al., 2021) and MoE (Shi et al., 2019; Sutter et al., 2020; Palumbo et al., 2023), where the joint distributions are approximated $q(\boldsymbol{z}|\mathbb{X}) = \prod_m q_m(\boldsymbol{z}|\boldsymbol{x}_m)$ and $q(\boldsymbol{z}|\mathbb{X}) = 1/M \sum_m q_m(\boldsymbol{z}|\boldsymbol{x}_m)$, respectively. However, none of these methods take into account the dependence between expert distributions, and failure to do so may harm the performance of multimodal VAEs. In addition, the MoE method introduces a subsampling of modalities that has a negative impact on the likelihood estimation and the generative quality of multimodal VAEs (Daunhawer et al., 2022). Instead, this research introduces an efficient consensus of experts method

---

[1]Sub-sampling in this research refers only to the definition of consensus distributions as a mixture model and the training of some, but not all, $\mathcal{L}_k$ terms (defined in Section 3).

to model the dependence between experts via errors of estimation.

# 3. Methods

We observe multimodal data $\mathbb{X}$ composed of $M$ modalities $\boldsymbol{x}_1, \boldsymbol{x}_2, \cdots, \boldsymbol{x}_M$ that are governed by a latent variable $\boldsymbol{z}$ through the conditional distribution $p(\mathbb{X}|\boldsymbol{z}) = p(\boldsymbol{x}_1|\boldsymbol{z})p(\boldsymbol{x}_2|\boldsymbol{z})\cdots p(\boldsymbol{x}_M|\boldsymbol{z})$. In this context, assuming a prior distribution $p(\boldsymbol{z})$, the marginal $p(\mathbb{X}) = \int p(\mathbb{X}|\boldsymbol{z})p(\boldsymbol{z})d\boldsymbol{z}$ and posterior $p(\boldsymbol{z}|\mathbb{X})$ distributions are intractable. Therefore, variational inference (VI) approximates the true and unknown posterior distribution $p(\boldsymbol{z}|\mathbb{X})$ with the parametric variational distribution $q(\boldsymbol{z}|\mathbb{X})$. It seems natural to minimize the Kullback-Leibler (KL) divergence $KL[q(\boldsymbol{z}|\mathbb{X})||p(\boldsymbol{z}|\mathbb{X})]$; however, it contains the intractable marginal distribution $KL[q(\boldsymbol{z}|\mathbb{X})||p(\boldsymbol{z}|\mathbb{X})] = \log p(\mathbb{X}) - \mathbb{E}_{q(\boldsymbol{z}|\mathbb{X})}[\log p(\mathbb{X}|\boldsymbol{z})] + KL[q(\boldsymbol{z}|\mathbb{X})||p(\boldsymbol{z})]$. Given that the KL divergence is strictly positive, it is straightforward to derive a lower bound on $\log p(\mathbb{X})$ as follows $\log p(\mathbb{X}) \geq \mathbb{E}_{q(\boldsymbol{z}|\mathbb{X})}[\log p(\mathbb{X}|\boldsymbol{z})] - KL[q(\boldsymbol{z}|\mathbb{X})||p(\boldsymbol{z})]$. However, we are interested in approximating the true posterior distribution $p(\boldsymbol{z}|\mathbb{X})$ even when only a subset $\mathbb{X}_k \in \mathcal{P}(\mathbb{X})$ is available. In the case of missing modalities, as shown by (Sutter et al., 2021), a valid lower bound for the available subset $\mathbb{X}_k \in \mathcal{P}(\mathbb{X})$ is

$$\mathcal{L}_k(\mathbb{X}) = \mathbb{E}_{q(\boldsymbol{z}|\mathbb{X}_k)}[\log p(\mathbb{X}|\boldsymbol{z})] - KL[q(\boldsymbol{z}|\mathbb{X}_k)||p(\boldsymbol{z})]. \quad (1)$$

To consider all scenarios of missing modalities, we optimize all lower bounds $\mathcal{L}_k(\mathbb{X})$ governed by its variational distribution $q(\boldsymbol{z}|\mathbb{X}_k)$. The next section introduces our proposed CoDE method for estimating joint variational posterior distributions taking into account the stochastic dependence between single-modality distributions. The main point to note is that the single-modality distributions have access to the same source of information. Therefore, assuming independence between them, as PoE and MoE do, is an overoptimistic assumption.

## 3.1. Consensus of Experts with Stochastic Dependence

In multimodal learning we are interested in generating latent variables $\boldsymbol{z}$ even when multimodal data $\mathbb{X}$ have missing modalities. Under this premise, there are $K = 2^M - 1$ distributions $q(\boldsymbol{z}|\mathbb{X}_k)$ to be learned given $M$ modalities, where $k = 1, 2, \cdots, K$ denotes a subset of the powerset $\mathcal{P}(\mathbb{X})$[2]. Since the number of distributions is exponential in the number of modalities, we need methods that scale linearly in $M$, more details in Appendix B . This research advances the consensus of experts method in (Winkler, 1981) by extending it to multivariate data, considering dependence between expert distributions, and applying it to multimodal

---

[2]In this research, we do not consider the empty set of $P(\mathbb{X})$ since there are no learnable parameters in $q(\boldsymbol{z})$.

VAEs. The CoDE method, therefore, represents the first principled Bayesian approach capable of approximating joint variational posterior distributions $q(\boldsymbol{z}|\mathbb{X}_k) \in \mathcal{P}(\mathbb{X})$ in multimodal VAEs.

**Definition 1.** All $M$ distributions $q(\boldsymbol{z}|\overline{\overline{\mathbb{X}}}_k = 1)$, where $\overline{\overline{\mathbb{X}}}_k$ denotes the cardinality of $\mathbb{X}_k$, are considered expert distributions, estimating the remaining unknown distributions $q(\boldsymbol{z}|\overline{\overline{\mathbb{X}}}_k > 1)$, which are called consensus distributions.

Definition 1 introduces expert and consensus distributions, also referred to as single-modality and joint posterior distributions in previous research. Each expert distribution provides information to estimate all consensus distributions in $\mathcal{P}(\mathbb{X})$. In the remainder of this section, we use standard Bayesian notation where $\boldsymbol{\theta}$ denotes unknown variables for which we compute posterior distributions.

**Definition 2.** Let $\boldsymbol{\theta}_k = (\theta_k^1, \theta_k^2, \cdots, \theta_k^D)^T$ denote the latent variable $\boldsymbol{z} \in \mathbb{R}^D$ for the $k$-th consensus distribution $q(\boldsymbol{z}|\mathbb{X}_k)$. Each $j$-th expert distribution provides a point estimate $\mu_j^d$ on the $d$-th dimension $\theta_k^d$, and the uncertainty about the estimation is expressed in the parameter $\sigma_j^d$ of each expert distribution. Therefore, the error of estimation of the $j$-th expert in the $d$-th dimension is $e_j^d = \mu_j^d - \theta_k^d$ and the error of estimation of all experts in the $d$-th dimension is $\boldsymbol{e}^d = (e_1^d, e_2^d, \cdots, e_{M'}^d)^T$, where $M'$ is the number of experts who evaluate the subset $\mathbb{X}_k$. The overall error of estimation on $\boldsymbol{\theta}_k$ is $\boldsymbol{e}_k = (\boldsymbol{e}^1, \boldsymbol{e}^2, \cdots, \boldsymbol{e}^D)^T$.

According to Definition 2, all expert distributions provide estimates of the unknown variable $\boldsymbol{\theta}$, as well as a measurement of the uncertainty on their estimation. Then, after considering the estimates of all relevant expert distributions, we can calculate the unknown consensus distributions. Note that only the consensus distribution corresponding to the subset with all elements, i.e. $q(\boldsymbol{z}|\overline{\overline{\mathbb{X}}} = M)$, contains estimates of all $M$ expert distributions. In general, consensus distributions are derived from different numbers of expert distributions, depending on the cardinality of the subset on which they are conditioned. Therefore, the size of the vector $\boldsymbol{e}_k$ depends on the cardinality of $\mathbb{X}_k$.

**Lemma 1.** *Assume that the error of estimation $\boldsymbol{e}_k$ of the $k$-th consensus distribution is a random variable with multivariate Gaussian distribution $\boldsymbol{e}_k \sim \mathcal{N}(\mathbf{0}, \boldsymbol{\Sigma}_k)$, where $\mathbf{0}$ is a $[M' \cdot D \times 1]$ vector, $M'$ is the number of experts who are elements of the subset $\mathbb{X}_k$, and $D$ is the dimensionality of $\boldsymbol{z}$. The covariance matrix is defined as*

$$\boldsymbol{\Sigma}_k = \begin{bmatrix} \boldsymbol{\Sigma}^1 & \mathbf{0} & \cdots & \mathbf{0} \\ \mathbf{0} & \boldsymbol{\Sigma}^2 & \cdots & \mathbf{0} \\ \vdots & \vdots & \ddots & \vdots \\ \mathbf{0} & \mathbf{0} & \cdots & \boldsymbol{\Sigma}^D \end{bmatrix},$$

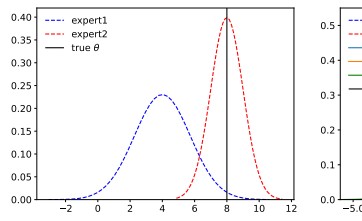
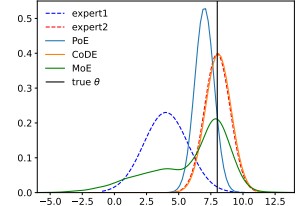

(a) Expert distributions    (b) Consensus distributions

*Figure 1.* Two univariate expert distributions with estimates $\mu_1 = 4$ and $\mu_2 = 8$ on the unknown variable $\theta = 8$. Expert 2 is more certain ($\sigma_2^2 = 1$) than expert 1 ($\sigma_1^2 = 3$).

*where*

$$\boldsymbol{\Sigma}^d = \begin{bmatrix} \sigma_1^2 & \sigma_{1,2} & \cdots & \sigma_{1,M'} \\ \sigma_{2,1} & \sigma_2^2 & \cdots & \sigma_{2,M'} \\ \vdots & \vdots & \ddots & \vdots \\ \sigma_{M',1} & \sigma_{M',2} & \cdots & \sigma_{M'}^2 \end{bmatrix},$$

*and $\mathbf{0}$ is a $[M' \times M']$ matrix. Let $\boldsymbol{\mu}^d = (\mu_1^d, \mu_2^d, \cdots, \mu_{M'}^d)^T$ be a vector with estimates of all expert distributions about the $d$-th dimension and $\boldsymbol{\mu}_k = (\boldsymbol{\mu}^1, \boldsymbol{\mu}^2, \cdots, \boldsymbol{\mu}^D)^T$ be the vector containing estimates of all expert distributions about all dimensions of $\boldsymbol{\theta}_k$. Therefore, the distribution of all estimates of the $k$-th consensus distributions is the multivariate Gaussian distribution $\boldsymbol{\mu}_k \sim \mathcal{N}(\boldsymbol{u}\boldsymbol{\theta}_k, \boldsymbol{\Sigma}_k)$, where $\boldsymbol{u}$ is a $[M' \cdot D \times D]$ design matrix with size $M'$ vectors of 1s along the diagonal and 0s elsewhere.*

There are a couple of interesting results according to Lemma 1 (see Appendix A.1 for the proof). First, the point estimates $\boldsymbol{\mu}_k$ provided by the expert distributions are random Gaussian variables. If $\mathcal{N}(\boldsymbol{u}\boldsymbol{\theta}_k, \boldsymbol{\Sigma}_k)$ is viewed as a function of $\boldsymbol{\theta}_k$ instead of $\boldsymbol{\mu}_k$, it can be interpreted as a likelihood function and we can calculate the posterior distribution of $\boldsymbol{\theta}_k$ in a principled Bayesian manner. Second, each covariance matrix $\boldsymbol{\Sigma}^d$ expresses both the uncertainty in the expert error of estimation on the $d$-th dimension of $\boldsymbol{z}$ (main diagonal) and the dependency between expert errors (off-diagonal). It seems reasonable to assume that expert distributions have overlapping information, or positive correlation, since the data modalities are simply different sources of information about the same object.

**Toy example:** To gain insight into the aggregation of expert distributions with CoDE, PoE, and MoE, Fig. 1 shows two experts with dependence on their assessment of the unknown parameter $\theta$. Experts 1 and 2 provide estimates $\mu_1 = 4$ and $\mu_2 = 8$, as well as a degree of uncertainty about their estimates $\sigma_1^2 = 3$ and $\sigma_2^2 = 1$. This information is *summarized* in expert distributions, assumed to be Gaussian. Taking estimates $(\mu_1, \mu_2)$ as samples from the likelihood function $\mathcal{N}(\boldsymbol{u}\theta, \boldsymbol{\Sigma})$, we can calculate the posterior (consensus) distribution of $\theta$. The dependence between experts is considered in the off-diagonal of $\boldsymbol{\Sigma}$ using

$\rho = 0.6$, $\sigma_1 = \sqrt{3}$, and $\sigma_2 = \sqrt{1}$. The estimated consensus distribution by CoDE recovers the unknown parameter $\theta$, by leaning even more towards expert 2. Note that MoE estimates a consensus distribution that is not sharper than any of the experts and does not recover, in expectation, the $\theta$ parameter. PoE underestimates the variance of the consensus distribution, as it neglects the dependence between experts. Consequently, the consensus distribution for PoE is sharper than that of any of the experts[3], even when it does not recover $\theta$. See Appendix B for additional details.

**Lemma 2.** *Assuming a flat prior distribution on the parameter $\boldsymbol{\theta}_k$ of the $k$-th consensus distribution and known covariance matrix $\boldsymbol{\Sigma}_k$, the posterior distribution is*

$$h(\boldsymbol{\theta}_k|\boldsymbol{\mu}_k) \sim \mathcal{N}(\boldsymbol{\mathcal{A}}_k^{-1}\boldsymbol{\mathcal{B}}_k, \boldsymbol{\mathcal{A}}_k^{-1}), \qquad (2)$$

*where $\boldsymbol{\mathcal{A}}_k = \boldsymbol{u}^T\boldsymbol{\Sigma}_k^{-1}\boldsymbol{u}$, $\boldsymbol{\mathcal{B}}_k = \boldsymbol{u}^T\boldsymbol{\Sigma}_k^{-1}\boldsymbol{\mu}_k$, $\boldsymbol{\Sigma}_k^{-1}$ is the inverse matrix of $\boldsymbol{\Sigma}_k$, and $\boldsymbol{u}$ and $\boldsymbol{\mu}_k$ are defined in Lemma 1.*

**Remark 1.** Lemma 2 subsumes PoE for $\boldsymbol{\Sigma}_k$ with diagonal matrices $\boldsymbol{\Sigma}^d$, in which case $h(\boldsymbol{\theta}_k|\boldsymbol{\mu}_k) \sim \mathcal{N}((\sum_i \mu_i \tau_i)(\sum_i \tau_i)^{-1}, (\sum_i \tau_i)^{-1})$. See Appendix B for details.

Lemma 2 (proof in Appendix A.2) shows that we can estimate the posterior distribution of the unknown parameter $\boldsymbol{\theta}_k$ after observing its estimates $\boldsymbol{\mu}_k$ from the expert distributions. Recall that Definition 2 refers to the latent variable $\boldsymbol{z}$ with $\boldsymbol{\theta}_k$, having $\boldsymbol{z} \sim q(\boldsymbol{z}|\mathbb{X}_k)$. Therefore, CoDE approximates consensus distributions in a principled Bayesian manner with the posterior distribution $q(\boldsymbol{z}|\mathbb{X}_k) \sim \mathcal{N}(\boldsymbol{\mathcal{A}}_k^{-1}\boldsymbol{\mathcal{B}}_k, \boldsymbol{\mathcal{A}}_k^{-1})$[4], and it can be used in any multimodal VAE that uses PoE as an approximation method. It is noteworthy that CoDE does not rely on sub-sampling techniques, which have been shown to harm the performance of multimodal VAEs (Daunhawer et al., 2022). Furthermore, note that choosing a diagonal matrix $\boldsymbol{\Sigma}^d$, as PoE does (see Remark 1), assumes that expert estimates are not correlated and, therefore, the variance of the consensus distribution is underestimated.

**Dependency between expert distributions:** Lemma 2 shows that the covariance matrix of the consensus distributions is a function of $\boldsymbol{\Sigma}_k$, which is composed of $\boldsymbol{\Sigma}^d$ on its main diagonal. The off-diagonal elements of $\boldsymbol{\Sigma}^d$ capture the dependency between the error of estimation of expert distributions. Therefore, in the forward pass the only unknown parameters in $\boldsymbol{\mathcal{A}}_k$ and $\boldsymbol{\mathcal{B}}_k$ to calculate the consensus distribution are the off-diagonal elements $\sigma_{j,i}$ in $\boldsymbol{\Sigma}^d$,

---

[3]Note that $\boldsymbol{\tau}_{PoE} = \boldsymbol{\tau}_1 + \cdots + \boldsymbol{\tau}_{M'}$, where $\boldsymbol{\tau}$ is the inverse of the variance, so $\boldsymbol{\tau}_{PoE} > \boldsymbol{\tau}_i$ or $\boldsymbol{\sigma}_{PoE}^2 < \boldsymbol{\sigma}_i^2$, for $i = 1, \cdots, M'$.

[4]Conditioning on $\boldsymbol{\mu}_k$ or $\mathbb{X}_k$ have the same meaning, as $\boldsymbol{\mu}_k$ are estimates from all experts in the subset $\mathbb{X}_k$.

as $(\mu_i, \sigma_i^2)$ are outputs of the encoder networks for all dimensions. We specify $\sigma_{i,j}$ as $\rho\sigma_j\sigma_i$ and the unknown correlation parameter $\rho$ is found by cross-validation over the values $[0, 0.2, 0.4, 0.6, 0.8]$. Note, CoDE does not impose any restriction on specifying $\Sigma^d$ differently, e.g. different or negative $\rho$ values, as long as it is an invertible matrix.

### 3.2. Evidence Lower Bound

To leverage the proposed CoDE for multi-modal VAEs, we follow an approach similar to that of (Sutter et al., 2021), where the objective function is the sum of all ELBOs $\mathcal{L}_k(\mathbb{X})$ in Eq. 1 that arises from all subsets $\mathbb{X}_k \in \mathcal{P}(\mathbb{X})$. In (Sutter et al., 2021) each $k$-th ELBO term contributes equally to optimization. In this work, however, we argue that if $\mathrm{tr}(\Sigma_j) > \mathrm{tr}(\Sigma_k)$, where $\mathrm{tr}(\cdot)$ is the trace of a matrix and $\Sigma$ is the covariance matrix of expert or consensus distributions, the distribution associated with the $k$-th subset is more confident in its estimate, indicating that it has relatively more reliable information than the $j$-th subset. Similarly, if $\overline{\overline{\mathbb{X}}}_j > \overline{\overline{\mathbb{X}}}_k$, the $j$-th subset contains relatively more information and it should contribute more to the optimization of the objective function. To avoid using equal weights in the weighted sum of all ELBOs, we introduce an indicator vector $\boldsymbol{\xi}$ that is drawn from the categorical distribution $\mathrm{Cat}(\boldsymbol{\pi})$, where $\boldsymbol{\pi} = (\pi_1, \cdots, \pi_K)$ is a probability vector, and $K$ is the number of subsets in $\mathcal{P}(\mathbb{X})$ (excluding the empty set). In our context, the indicator vector $\boldsymbol{\xi}$ maps each $k$-th subset $\mathbb{X}_k \in \mathcal{P}(\mathbb{X})$ to either 1 or 0, and each event occurs with probability $\pi_k$. For example, the first subset in $\mathcal{P}(\mathbb{X})$ occurs with probability $Pr(\boldsymbol{\xi}|\mathbb{X}_1) = \pi_1$, where $\boldsymbol{\xi} = [1, 0, \cdots, 0]$.

Assuming the generative model $p(\mathbb{X}, \boldsymbol{z}, \boldsymbol{\xi}) = p(\mathbb{X}|\boldsymbol{z})p(\boldsymbol{z})p(\boldsymbol{\xi})$, where $p(\boldsymbol{\xi}) \sim \mathrm{Cat}(\boldsymbol{\eta})$ is a prior categorical distribution with equal probabilities $\boldsymbol{\eta} = (1/K, \cdots, 1/K)$, and the inference model $q(\boldsymbol{z}, \boldsymbol{\xi}|\mathbb{X}_k) = q(\boldsymbol{z}|\mathbb{X}_k)q(\boldsymbol{\xi}|\mathbb{X}_k)$, where $q(\boldsymbol{\xi}|\mathbb{X}_k) \sim \mathrm{Cat}(\boldsymbol{\pi})$ is the posterior distributions of $\boldsymbol{\xi}$, the variational lower bound of the CoDE-VAE model for a single data point is given in Lemma 3.

**Lemma 3.** *The concave function*

$$\mathcal{L}(\mathbb{X}) = \sum_{\mathbb{X}_k} \left\{ \pi_k \left[ \mathbb{E}_{q_\phi(\boldsymbol{z}|\mathbb{X}_k)}[\log p_{\boldsymbol{\theta}}(\mathbb{X}|\boldsymbol{z})] \right. \right.$$
$$\left. \left. - KL[q_\phi(\boldsymbol{z}|\mathbb{X}_k)||p(\boldsymbol{z})] \right] + \mathcal{H}(q_\phi(\boldsymbol{\xi}|\mathbb{X}_k)) \right\} + C, \quad (3)$$

*where $\mathcal{H}$ denotes the entropy function and $C$ is a constant term, is a variational lower bound on $\log p(\mathbb{X})$ that optimizes all $k$-th ELBOs $\mathcal{L}_k$ with weight coefficients $\pi_k$.*

The ELBO of the CoDE-VAE model in Eq. 3 (proof in Appendix A.3) parameterizes the conditional likelihoods $p_{\boldsymbol{\theta}}(\mathbb{X}|\boldsymbol{z})$, expert and consensus distributions $q_\phi(\boldsymbol{z}|\mathbb{X}_k)$ with neural networks, where $\boldsymbol{\theta}$ and $\phi$ denote the learnable

weights of the neural networks. The weight coefficients $\boldsymbol{\pi}$ are learnable parameters optimized by entropy maximization. See Algorithm 1 in Appendix D.1 for details.

## 4. Experiments

Following the standard experimental setup in this domain (Sutter et al., 2021; Daunhawer et al., 2022; Palumbo et al., 2023), performance is evaluated using the following multimodal datasets: MNIST-SVHN-Text dataset (Sutter et al., 2020) composed of matching MNIST and SVHN digits and a text describing the digit; PolyMNIST (Sutter et al., 2021) composed of 5 MNIST images of the same digit, but different background and handwriting style; and The Caltech Birds (CUB) dataset (Wah et al., 2011; Daunhawer et al., 2022), which is composed of images of birds paired with captions describing each bird. This is a significantly more challenging version compared to the one used in (Shi et al., 2019), where they use ResNet embeddings rather than actual images making the learning task significantly easier. Note that these datasets have different levels of complexity due to the shared and modality-specific information across modalities. MNIST-SVHN-Text and CUB have a large amount of modality-specific information given the heterogeneity across modalities. On the other hand, PolyMNIST contains a moderate amount of modality-specific information, but is a suitable dataset for analyzing the quality gap as the number of input modalities for model training increases.

We compare the performance of the CoDE-VAE model[5] with that of MVAE (Wu & Goodman, 2018), MMVAE (Shi et al., 2019), mmJSD (Sutter et al., 2020), MoPoE (Sutter et al., 2021), MVTCAE (Hwang et al., 2021), and MMVAE+ (Palumbo et al., 2023). Note that MMVAE+ is the only model in the comparison that utilizes modality-specific and shared latent variables in the decoder to improve the quality of the generated modalities. We measure the performance of all models in terms of the generative coherence and the generative quality of the generated modalities conditioned on both latent variables from the joint posterior and prior distributions. In addition, to assess the quality of the approximation of joint posterior distributions, we measure the tightness of the lower bound by log-likelihood estimations. Finally, for completeness, the discriminative power of the joint latent representations $\boldsymbol{z}$ using linear classifiers is shown in the appendix. Limitations of our proposed model are provided in Appendix C, while details on datasets, evaluation criteria, network architectures, training procedure, and extra results are in Appendix D.

All experiments report the average performance over three different runs, and the hyperparameters $\beta$ (Higgins et al.,

---

[5] https://github.com/rogelioamancisidor/codevae.

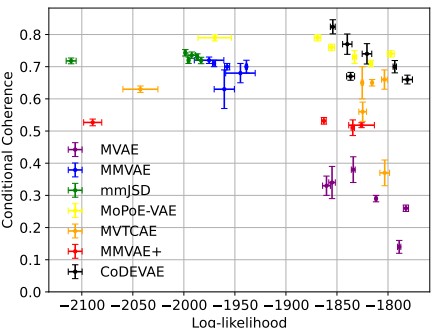 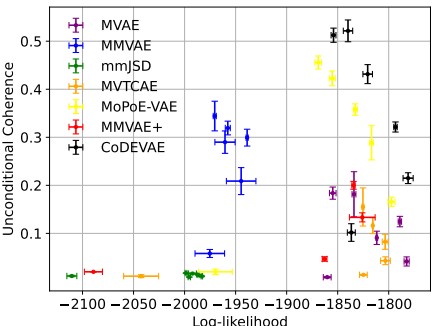

*Figure 2.* Trade-off between generative coherence (↑) and joint log-likelihoods (↑) on the MNIST-SVHN-Text test set.

*Table 1.* Generative quality measured by FID scores (↓) and generative coherence (↑) on the MNIST-SVHN-Text test set.

|  | Conditional Coherence | Unconditional Coherence | Conditional FID | Unconditional FID |
|---|---|---|---|---|
| MVAE | 0.38 ±0.042 | 0.18 ±0.047 | 41.09 ±2.57 | **44.96** ±3.26 |
| MMVAE | 0.72 ±0.014 | 0.34 ±0.031 | 153.96 ±7.73 | 111.72 ±5.07 |
| mmJSD | 0.74 ±0.013 | 0.02 ±0.001 | 148.47 ±4.38 | 291.59 ±1.38 |
| MoPoE-VAE | 0.79 ±0.015 | 0.46 ±0.014 | 105.36 ±1.49 | 101.43 ±7.49 |
| MVTCAE | 0.66 ±0.031 | 0.15 ±0.040 | **38.34** ±1.85 | 52.86 ±0.48 |
| MMVAE+ | 0.53 ±0.012 | 0.20 ±0.005 | 54.69 ±0.86 | 77.35 ±1.37 |
| CoDEVAE | **0.82** ±0.022 | **0.51** ±0.015 | 79.11 ±2.73 | 71.40 ±1.93 |

2016) and $\rho$ are found by cross-validation over the values $[0.1, 1, 5, 10, 15, 20]$ and $[0, 0.2, 0.4, 0.6, 0.8]$, respectively. In addition, we consider $\beta = 2.5$ in the experiments on the PolyMNIST data to be consistent with the setup in (Palumbo et al., 2023). Given that the parameter $\rho$ is exclusive to CoDE-VAE, all experiments present results that correspond to the $\rho$ value demonstrating the strongest overall performance. We compare the performance of all models across the entire range of $\beta$ values. When only single values are reported, e.g., in tables or figures, the results correspond to the optimal $\beta$ for each method to ensure a fair comparison as different models achieve optimal performance at different values (Daunhawer et al., 2022).

### 4.1. MNIST-SVHN-Text

An important aspect of multimodal VAEs is their ability to approximate the joint likelihood of multimodal data without sacrificing generative coherence. Fig. 2 shows the trade-off between generative coherence and log-likelihood estimation for all $\beta$ values. Simple models based on MoE, e.g. mmJSD and MMVAE, show poor log-likelihoods, which is a result of modality sub-sampling and clearly prevents a tight approximation as there is a large amount of modality-specific information in the data. Even though MMVAE+ approximates the consensus distributions with the MoE, its log-likelihood estimations are tighter due to the factorization of the latent space that learns the modality-specific information. On the other hand, methods based on PoE all have relatively tight log-likelihood estimations.

However, simple models such as MVAE, trades off tighter log-likelihoods for generative coherence. The CoDE-VAE models finds a balance between generative coherence and the approximation of the joint likelihood, showing superior performance than all benchmark models for all beta values.

While generative coherence is important, the generated modalities should be of a high-quality. Table 1 compares generative coherence with generative quality measured by FID scores (Heusel et al., 2018). Both the MVAE and MVTCAE models are able to generate high-quality image modalities, but this is accomplished by compromising generative coherence. In the family of models using MoE, MMVAE+ achieves the highest conditional quality, which is an obvious result of the modality-specific variables. However, this is achieved by trading off generative coherence. On the other hand, the CoDE-VAE model is able to generate highly coherent images without too much compromise of the generative quality.

### 4.2. PolyMNIST

**Generative quality gap:** The generative quality gap (Daunhawer et al., 2022) is defined as the difference measured by FID scores between modalities generated with unimodal VAEs and multimodal VAEs. Methods that rely on sub-sampling of modalities suffer from it and the gap increases with the number of modalities used for model training (Daunhawer et al., 2022). To assess the generative quality gap, we train all models with 2-5 modalities in the PolyMNIST data, always using the modalities $m_0$ and

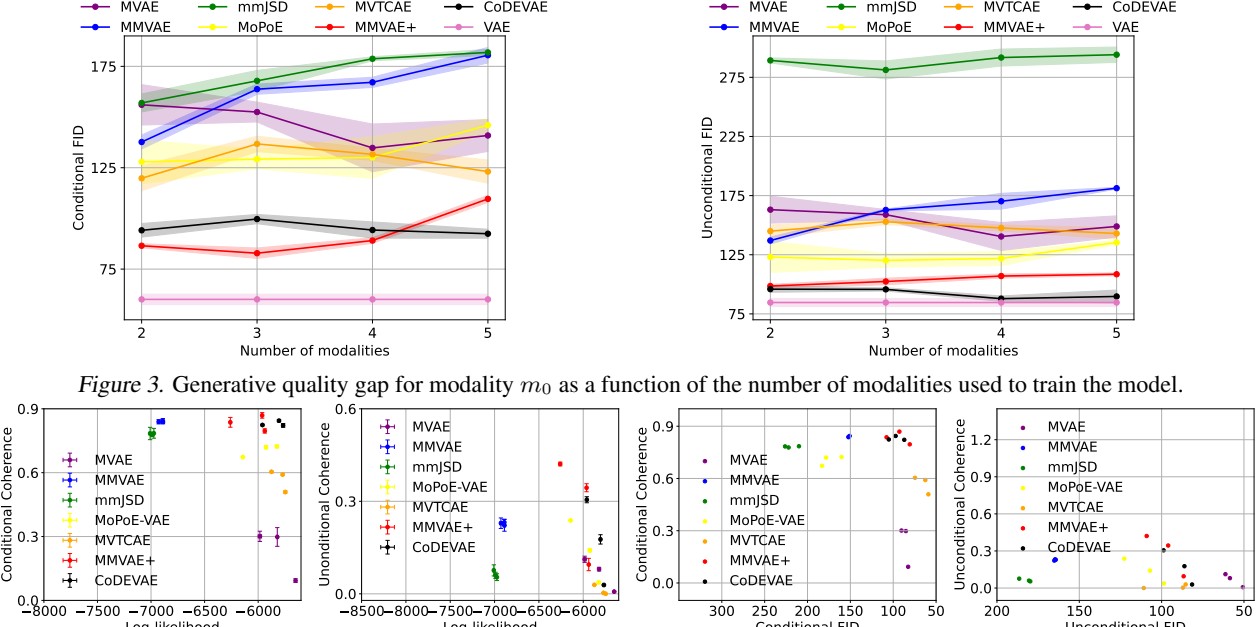

*Figure 3.* Generative quality gap for modality $m_0$ as a function of the number of modalities used to train the model.

*Figure 4.* Trade-off between generative coherence (↑) and log-likelihood estimation (↑), and between generative coherence and generative quality (↓) for $\beta \in [1, 2.5, 5]$ on the PolyMNIST dataset.

$m_1$ and generating modality $m_0$. For each of these four training scenarios, Fig. 3 shows that the generative quality gap increases with the number of modalities in all models using sub-sampling of modalities. Only the MVAE, MVTCAE, and CoDE-VAE models, which do not rely on sub-sampling of modalities and MoE, have a linear or decreasing trend as the number of modalities increases. Only the CoDE-VAE model achieves the same unconditional generative quality as unimodal VAEs when the number of modalities is increased to 4 and 5.

**Log-likelihoods, generative quality, and coherence:** Fig. 4 shows the same pattern as for MNIST-SVHN-Text, where the MVAE and mmJSD models have a poor approximation of the joint distribution of the data. However, due to the moderate amount of modality-specific information in PolyMNST, the generative coherence for these models is not compromised, specially conditional coherence. We can corroborate that models using the PoE to approximate consensus distributions have higher log-likelihoods estimations. However, models like MVAE and MVTCAE trade off generative coherence. Both the MMVAE+ and CoDE-VAE models are able to balance the trade-off between generative coherence and log-likelihoods estimations.

The pattern in generative quality is similar to that in the log-likelihoods estimation, except for the relatively poor FID scores achieve by the MoPoE model, specially in the conditional scenario. Both MVAE and MVTCAE generate high-quality modalities but at the expense of generative coherence, which is clearly not desirable. Similarly

to the log-likelihood estimations, MMVAE+ and CoDE-VAE find a balance in the generative quality and generative coherence of modalities. However, we would like to emphasize that MMVAE+ is not able to achieve significantly better FID scores than CoDE-VAE despite using modality-specific variables that improve the generative model. It is noteworthy that MMVAE+ shows higher coherence due to the low $\beta$ values in this experiment, which replicate the setup in (Palumbo et al., 2023). CoDE-VAE performs better at higher $\beta$ values, achieving coherence of 0.38 at $\beta = 10$ for example.

### 4.3. CUB

To validate the performance of our proposed model on complex real-world data, we test the generative coherence and generative quality on the CUB data. Table 2 shows the quantitative results for the caption-to-image generative quality and coherence, while the qualitative results can be found in Appendix D.5. Conditional coherence is calculated using the approach introduced in (Palumbo et al., 2023), as the captions describing the CUB data do not necessarily describe the shared content between modalities. The results obtained by MVAE, mmJSD, MoPoE, and MVTCAE reflect the complexity of the CUB data when real images are considered. While MMVAE is able to obtain relatively high coherence, its performance totally depends on the estimation of the variance parameter in the prior distribution. CoDE-VAE achieves relative high coherence without compromising the quality of the generated images, which is not far away from MMVAE+ that uses

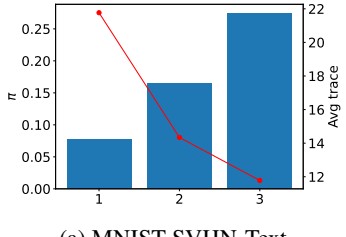

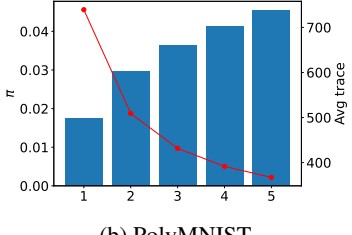

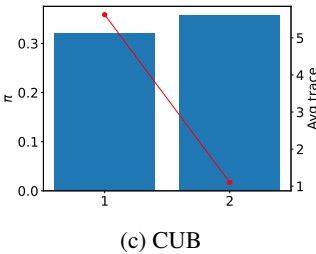

(a) MNIST-SVHN-Text  (b) PolyMNIST  (c) CUB

*Figure 5.* The bars show the learned coefficients $\pi_k$ (left axis), while red lines show the average trace of the covariance matrix of the distribution $q(\boldsymbol{z}|\mathbb{X}_k)$ (right axis). All values are averages over the subsets with the same cardinality.

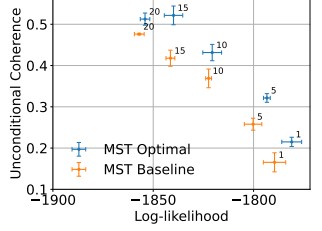

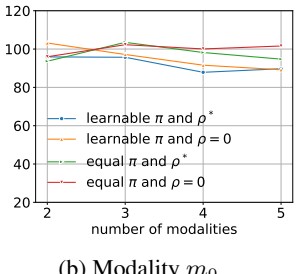

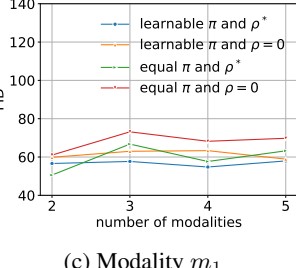

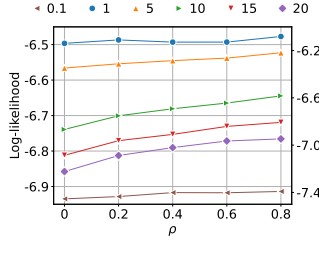

(a) Optimal vs Baseline.  (b) Modality $m_0$  (c) Modality $m_1$  (d) Log-likelihoods

*Figure 6.* a) Contribution of learning $\pi_k$ and cross-validating $\rho$ ($\beta$ values at the top); b) and c): Effect of the coefficients $\pi_k$ and correlation $\rho$ on FID scores on PolyMNIST; d): Log-likelihoods (in thousands) for different $\beta$'s as a function of $\rho$ ($\beta = 0.1$ on the right axis).

*Table 2.* Caption-to-image conditional generative quality and generative coherence on the CUB dataset.

|  | Conditional Coherence | Conditional FID |
|---|---|---|
| MVAE | 0.27 ±0.007 | 172.21 ±39.61 |
| MMVAE | 0.71 ±0.057 | 232.20 ±2.14 |
| mmJSD | 0.56 ±0.158 | 262.80 ±6.93 |
| MoPoE-VAE | 0.58 ±0.158 | 265.55 ±4.01 |
| MVTCAE | 0.22 ±0.007 | 208.43 ±1.10 |
| MMVAE+ | 0.72 ±0.090 | **164.94** ±1.50 |
| CoDEVAE | **0.75** ±0.050 | 175.97 ±0.30 |

modality-specific variables to improve precisely this metric.

### 4.4. Contribution of each $\mathcal{L}_k$ to optimization

Fig. 5a shows the learned coefficients $\pi_k$ (left axis) associated with each $k$-th ELBO $\mathcal{L}_k(\mathbb{X})$ in the overall ELBO in Eq. 3, as well as the average trace (right axis) of the covariance matrix of the $k$-th distribution. In both cases, averaged over subsets with equal cardinality. The average $\pi_k$ for subsets $\overline{\overline{\mathbb{X}}}_k = 1$ is lower than that of subsets $\overline{\overline{\mathbb{X}}}_k \geq 2$, and the ELBO that dominates optimization on the MNIST-SVHN-Text data is $\overline{\overline{\mathbb{X}}} = 3$. On the other hand, as the average trace increases, meaning that the distributions are relatively uncertain about their estimates, the average weight $\pi_k$ decreases. Similarly, Fig. 5b and 5c show that for the PolyMNIST and CUB datasets, subsets with more modalities contribute more to optimization and have smaller average trace values, indicating decreased uncertainty as modalities increase. The findings validate our motivation to learn each

$k$-th ELBO term's contribution to optimization, since distributions for which experts are certain should have a greater contribution.

### 4.5. Ablation Experiments: the effect of $\pi$ and $\rho$

Fig. 6a shows that considering dependencies between experts and learning $\pi_k$ yields higher performance across all beta values (blue error bars), compared to assuming equal weights and independent expert distributions (orange error bars), using the MNIST-SVHN-Text dataset. See Appendix F for results on PolyMNIST. Furthermore, we train the CoDE-VAE model optimizing Eq. 3 and using 2-5 modalities of the PolyMNIST data in the following scenarios: learning the weights $\pi$ and i) cross-validating $\rho$ (denoted as $\rho^*$) or ii) using $\rho = 0$; alternatively using equal weights $\pi_k$ for all ELBOs and iii) cross-validating $\rho$ or iv) using $\rho = 0$. Figs. 6b and 6c reveal that learning $\boldsymbol{\pi}$ and considering dependence among experts improve FID scores for the generated modalities. Finally, Fig. 6d shows that average joint log-likelihoods increase with higher $\rho$ and lower $\beta$ values.

## 5. Conclusion

This research introduces CoDE, a novel consensus of experts method that, unlike existing approaches, takes into account the stochastic dependence between expert distributions through the experts' error of estimation. Based on CoDE, we introduce CoDE-VAE, an innovative multimodal VAE that optimizes each ELBO associated with every $k$-th subset within $\mathcal{P}(\mathbb{X})$ by learning its contribution

$\pi_k$. Extensive experimentation shows that CoDE-VAE exhibits better performance in terms of balancing the trade-off between generative coherence and generative quality, as well as generating more precise log-likelihood estimations. As the CoDE-VAE optimization does not rely on sub-sampling of modalities, it reduces the generative quality gap as modalities increase, which is a desirable property that is missing in most current methods. Furthermore, CoDE-VAE achieves a classification accuracy comparable to that of current models. Finally, ablation experiments show the benefit of modeling the dependence between expert distribution and learning the contribution of each ELBO term to optimization.

## Acknowledgments

RJ, SY and MK are affiliated with *Visual Intelligence*, a Centre for Research-based Innovation funded by the Research Council of Norway (RCN) and consortium partners, grant no. 309439. RJ is affiliated with Pioneer Centre for AI, DNRF grant number P1. The work was further supported by RCN FRIPRO grant no. 315029 and RCN IKTPLUSS grant no. 303514. Finally, we acknowledge Sigma2 (Norway) for awarding this project access to the LUMI supercomputer, owned by the EuroHPC Joint Undertaking, hosted by CSC (Finland) and the LUMI consortium through project no. NN10040K.

## Impact Statement

This research presents work whose main objective is to advance the field of multimodal VAEs by introducing a consensus of experts method, which leverages the dependence between expert (single-modality) distributions, to approximate consensus (joint posterior) distributions of any combination of different modalities. This is not a trivial task and the existing literature usually assumes the independence between expert distributions for simplicity, which is an over-optimistic assumption in practice. Although the focus of our work is on multimodal VAEs, our proposed *consensus of dependent experts* (CoDE) method has a broader impact, as previous research in other domains has aggregated information from expert distributions, such as natural language processing to combine distributions of hidden Markov models (Brown & Hinton, 2001), image synthesis to combine modalities and generate images using generative adversarial networks (Huang et al., 2022), large language models for comparative assessment of texts (Liusie et al., 2024), in early-exit ensembles (Allingham & Nalisnick, 2022), or in diffusion models to aggregate distillation of diffusion policies (Zhou et al., 2024). There are many potential societal consequences of our work, none of which we feel must be specifically highlighted here.

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

# Appendix

## A. Proofs

### A.1. PROOF OF LEMMA 1

*Proof.* Let us rewrite the error of estimation as $\boldsymbol{e}_k = (\boldsymbol{\mu}_k - \boldsymbol{u}\boldsymbol{\theta}_k)$, where $\boldsymbol{u}$ is a $[M' \cdot D \times D]$ design matrix with size $M'$ vectors of 1s along the diagonal and 0s elsewhere. Furthermore, assume that the error of estimation has a Gaussian distribution $\boldsymbol{e}_k \sim \mathcal{N}(\boldsymbol{0}, \boldsymbol{\Sigma}_k)$. Therefore, the expectation of $\boldsymbol{\mu}_k$ is

$$
\begin{aligned}
\mathbb{E}[\boldsymbol{\mu}_k] &= \mathbb{E}[\boldsymbol{e}_k + \boldsymbol{u}\boldsymbol{\theta}_k] \\
&= \mathbb{E}[\boldsymbol{e}_k] + \boldsymbol{u}\boldsymbol{\theta}_k \\
&= \boldsymbol{u}\boldsymbol{\theta}_k
\end{aligned}
\tag{4}
$$

Similarly, the covariance matrix is

$$
\begin{aligned}
\mathbb{E}[(\boldsymbol{\mu}_k - \mathbb{E}[\boldsymbol{\mu}_k])(\boldsymbol{\mu}_k - \mathbb{E}[\boldsymbol{\mu}_k])^T] &= \mathbb{E}[(\boldsymbol{e}_k + \boldsymbol{u}\boldsymbol{\theta}_k - \mathbb{E}[\boldsymbol{\mu}_k])(\boldsymbol{e}_k + \boldsymbol{u}\boldsymbol{\theta}_k - \mathbb{E}[\boldsymbol{\mu}_k])^T] \\
&= \mathbb{E}[(\boldsymbol{e}_k - \mathbb{E}[\boldsymbol{e}_k])(\boldsymbol{e}_k - \mathbb{E}[\boldsymbol{e}_k])^T] \\
&= \boldsymbol{\Sigma}_k.
\end{aligned}
\tag{5}
$$

Given that $\boldsymbol{\mu}_k$ is a linear function of a random variable with a multivariate Gaussian distribution, it follows that $\boldsymbol{\mu}_k$ is also multivariate Gaussian, i.e, $\boldsymbol{\mu}_k \sim \mathcal{N}(\boldsymbol{u}\boldsymbol{\theta}_k, \boldsymbol{\Sigma}_k)$. $\square$

### A.2. PROOF OF LEMMA 2

*Proof.* Assuming an improper flat prior distribution on $\boldsymbol{\theta}_k$, and dropping the subscript $k$ to avoid cluttering the notation, the posterior distribution of the $k$-th consensus distribution is proportional to the likelihood function, i.e. $h(\boldsymbol{\theta}|\boldsymbol{\mu}) \propto \mathcal{N}(\boldsymbol{u}\boldsymbol{\theta}_k, \boldsymbol{\Sigma}_k)$. Therefore,

$$
\begin{aligned}
h(\boldsymbol{\theta}|\boldsymbol{\mu}) &\propto \exp\left[-\frac{1}{2}[(\boldsymbol{\mu} - \boldsymbol{u}\boldsymbol{\theta})^T \boldsymbol{\Sigma}^{-1}(\boldsymbol{\mu} - \boldsymbol{u}\boldsymbol{\theta})]\right] \\
&= \exp\left[-\frac{1}{2}[\boldsymbol{\mu}^T \boldsymbol{\Sigma}^{-1}\boldsymbol{\mu} - \boldsymbol{\mu}^T \boldsymbol{\Sigma}^{-1}\boldsymbol{u}\boldsymbol{\theta} - \boldsymbol{\theta}^T \boldsymbol{u}^T \boldsymbol{\Sigma}^{-1}\boldsymbol{\mu} + \boldsymbol{\theta}^T \boldsymbol{u}^T \boldsymbol{\Sigma}^{-1}\boldsymbol{u}\boldsymbol{\theta}]\right] \\
&= \exp\left[-\frac{1}{2}[\boldsymbol{\theta}^T \boldsymbol{\mathcal{A}}\boldsymbol{\theta} - 2\boldsymbol{\theta}^T \boldsymbol{\mathcal{B}} + \mathcal{C}]\right] \\
&= \exp\left[-\frac{1}{2}[\boldsymbol{\theta}^T \boldsymbol{\mathcal{A}}\boldsymbol{\mathcal{A}}\boldsymbol{\mathcal{A}}^{-1}\boldsymbol{\theta} - 2\boldsymbol{\theta}^T \boldsymbol{\mathcal{A}}\boldsymbol{\mathcal{A}}^{-1}\boldsymbol{\mathcal{B}} + \boldsymbol{\mathcal{B}}^T \boldsymbol{\mathcal{A}}^{-1}\boldsymbol{\mathcal{B}} - \boldsymbol{\mathcal{B}}^T \boldsymbol{\mathcal{A}}^{-1}\boldsymbol{\mathcal{B}} + \mathcal{C}]\right] \\
&\propto \exp\left[-\frac{1}{2}[(\boldsymbol{\mathcal{A}}\boldsymbol{\theta} - \boldsymbol{\mathcal{B}})^T \boldsymbol{\mathcal{A}}^{-1}(\boldsymbol{\mathcal{A}}\boldsymbol{\theta} - \boldsymbol{\mathcal{B}})]\right],
\end{aligned}
$$

which is an exponential form on $\boldsymbol{\theta}$. Taking the first derivative of $\log h(\boldsymbol{\theta}|\boldsymbol{\mu})$ wrt $\boldsymbol{\theta}$ and solving for $\boldsymbol{\theta}$, we obtain the expectation of the posterior distribution.

$$
\frac{\partial}{\partial \boldsymbol{\theta}} \log h(\boldsymbol{\theta}|\boldsymbol{\mu}) = -\boldsymbol{\theta}\boldsymbol{\mathcal{A}} + \boldsymbol{\mathcal{B}}.
$$

Similarly, the variance of the posterior distribution is given at $-\left(\frac{\partial^2}{\partial \boldsymbol{\theta}^2} \log h(\boldsymbol{\theta}|\boldsymbol{\mu})\right)^{-1}$.

$$
\frac{\partial^2}{\partial \boldsymbol{\theta}^2} \log h(\boldsymbol{\theta}|\boldsymbol{\mu}) = -\boldsymbol{\mathcal{A}}.
$$

Therefore, the posterior distribution

$$
h(\boldsymbol{\theta}|\boldsymbol{\mu}) \sim \mathcal{N}(\boldsymbol{\mathcal{A}}^{-1}\boldsymbol{\mathcal{B}}, \boldsymbol{\mathcal{A}}^{-1})
\tag{6}
$$

is also Gaussian, where $\boldsymbol{\mathcal{A}} = \boldsymbol{u}^T \boldsymbol{\Sigma}^{-1}\boldsymbol{u}$, $\boldsymbol{\mathcal{B}} = \boldsymbol{u}^T \boldsymbol{\Sigma}^{-1}\boldsymbol{\mu}$, $\boldsymbol{\Sigma}^{-1}$ is the inverse matrix of $\boldsymbol{\Sigma}$, $\boldsymbol{u}$ is a $[M' \cdot D \times D]$ design matrix with size $M'$ vectors of 1s along the diagonal and 0s elsewhere, and $\boldsymbol{\mu}$ is a $[M' \cdot D \times 1]$ vector with point estimates from relevant expert distributions on all D dimensions of the $k$-th consensus distribution. $\square$

A.3. PROOF OF LEMMA 3

*Proof.* We introduce an indicator vector $\boldsymbol{\xi}$ that maps each $k$-th subset $\mathbb{X}_k \in \mathcal{P}(\mathbb{X})$ to 1 or 0. We assume that both prior and posterior distributions of the indicator vector are categorical, i.e. $\boldsymbol{\xi} \sim \text{Cat}(\boldsymbol{\pi})$. Therefore, the probability that the subset $\mathbb{X}_k$ occurs is $\pi_k$, where $\pi_k \in \boldsymbol{\pi}$. Furthermore, we assume the inference model $q(\boldsymbol{z}, \boldsymbol{\xi} | \mathbb{X}_k) = q(\boldsymbol{z} | \mathbb{X}_k) q(\boldsymbol{\xi} | \mathbb{X}_k)$, where $q(\boldsymbol{z} | \mathbb{X}_k)$ and $q(\boldsymbol{\xi} | \mathbb{X}_k)$ are variational distributions, which are conditionally independent given $\mathbb{X}_k$.

Minimizing the sum of all the Kullback-Leibler divergences between the intractable posterior distribution $p(\boldsymbol{z}, \boldsymbol{\xi} | \mathbb{X}) = \frac{p(\mathbb{X} | \boldsymbol{z}) p(\boldsymbol{z}) p(\boldsymbol{\xi})}{p(\mathbb{X})}$ and the inference model, we obtain the following.

$$\sum_{\mathbb{X}_k} KL[q(\boldsymbol{z}, \boldsymbol{\xi} | \mathbb{X}_k) || p(\boldsymbol{z}, \boldsymbol{\xi} | \mathbb{X})] = \sum_{\mathbb{X}_k} \mathbb{E}_{q(\boldsymbol{z}, \boldsymbol{\xi} | \mathbb{X}_k)} \left[ \log \frac{q(\boldsymbol{z}, \boldsymbol{\xi} | \mathbb{X}_k)}{p(\mathbb{X} | \boldsymbol{z}) p(\boldsymbol{z}) p(\boldsymbol{\xi})} + \log p(\mathbb{X}) \right]$$

$$= \sum_{\mathbb{X}_k} \mathbb{E}_{q(\boldsymbol{z}, \boldsymbol{\xi} | \mathbb{X}_k)} \left[ \log \frac{q(\boldsymbol{z}, \boldsymbol{\xi} | \mathbb{X}_k)}{p(\mathbb{X} | \boldsymbol{z}) p(\boldsymbol{z}) p(\boldsymbol{\xi})} \right] + (2^M - 1) \log p(\mathbb{X})$$

$$\propto \sum_{\mathbb{X}_k} \mathbb{E}_{q(\boldsymbol{z}, \boldsymbol{\xi} | \mathbb{X}_k)} \left[ \log \frac{q(\boldsymbol{z}, \boldsymbol{\xi} | \mathbb{X}_k)}{p(\mathbb{X} | \boldsymbol{z}) p(\boldsymbol{z}) p(\boldsymbol{\xi})} \right] + \log p(\mathbb{X}),$$

where in line 3 we factor out $2^M - 1$ as a constant, which should not affect the optimization (Wu & Goodman, 2019). Assuming a categorical prior distribution for $p(\boldsymbol{\xi})$ with probability mass function $\boldsymbol{\eta} = (1/K, \cdots, 1/K)$, where $K = 2^M - 1$, i.e. equal prior probabilities for all subsets, we obtain

$$\log p(\mathbb{X}) \propto \sum_{\mathbb{X}_k} \mathbb{E}_{q(\boldsymbol{z}, \boldsymbol{\xi} | \mathbb{X}_k)} \left[ \log \frac{p(\mathbb{X} | \boldsymbol{z}) p(\boldsymbol{z}) p(\boldsymbol{\xi})}{q(\boldsymbol{z} | \mathbb{X}_k) q(\boldsymbol{\xi} | \mathbb{X}_k)} \right] + \sum_{\mathbb{X}_k} KL[q(\boldsymbol{z}, \boldsymbol{\xi} | \mathbb{X}_k) || p(\boldsymbol{z}, \boldsymbol{\xi} | \mathbb{X})]$$

$$\log p(\mathbb{X}) \geq \sum_{\mathbb{X}_k} \mathbb{E}_{q(\boldsymbol{z}, \boldsymbol{\xi} | \mathbb{X}_k)} \left[ \log \frac{p(\mathbb{X} | \boldsymbol{z}) p(\boldsymbol{z}) p(\boldsymbol{\xi})}{q(\boldsymbol{z} | \mathbb{X}_k) q(\boldsymbol{\xi} | \mathbb{X}_k)} \right]$$

$$= \sum_{\mathbb{X}_k} \mathbb{E}_{q(\boldsymbol{\xi} | \mathbb{X}_k)} \mathbb{E}_{q(\boldsymbol{z} | \mathbb{X}_k)} \left[ \log p(\mathbb{X} | \boldsymbol{z}) + \log p(\boldsymbol{z}) - \log q(\boldsymbol{z} | \mathbb{X}_k) - \log q(\boldsymbol{\xi} | \mathbb{X}_k) \right]$$

$$+ \sum_{\mathbb{X}_k} \mathbb{E}_{q(\boldsymbol{\xi} | \mathbb{X}_k)} [\log p(\boldsymbol{\xi})]$$

$$= \sum_{\mathbb{X}_k} \left[ \mathbb{E}_{q(\boldsymbol{\xi} | \mathbb{X}_k)} \mathbb{E}_{q(\boldsymbol{z} | \mathbb{X}_k)} [\log p(\mathbb{X} | \boldsymbol{z})] - KL[q(\boldsymbol{z} | \mathbb{X}_k) || p(\boldsymbol{z})] + \mathcal{H}(q(\boldsymbol{\xi} | \mathbb{X}_k)) \right]$$

$$+ \sum_{\mathbb{X}_k} \mathbb{E}_{q(\boldsymbol{\xi} | \mathbb{X}_k)} [\log p(\boldsymbol{\xi})],$$

Note that the expected value of the indicator variable $\boldsymbol{\xi}$ iif the $k$-th subset occurs is

$$\mathbb{E}[\boldsymbol{\xi} | \mathbb{X}_k] = \boldsymbol{\xi} \cdot \boldsymbol{\pi}$$
$$= [0, \cdots, 1_k, \cdots, 0] \cdot [\pi_1, \cdots, \pi_k, \cdots]$$
$$= \pi_k,$$

(Cormen et al., 2022) (Lemma 5.1 p. 130), and that $\sum_{\mathbb{X}_k} \mathbb{E}_{q(\boldsymbol{\xi} | \mathbb{X}_k)} [\log p(\boldsymbol{\xi})] = \log p(\boldsymbol{\xi})$ is just a constant term[6]. Therefore, the variational lower bound for a single data point is

$$\log p(\mathbb{X}) \geq \sum_{\mathbb{X}_k} \left\{ \pi_k \left[ \mathbb{E}_{q_{\boldsymbol{\phi}}(\boldsymbol{z} | \mathbb{X}_k)} [\log p_{\boldsymbol{\theta}}(\mathbb{X} | \boldsymbol{z})] - KL[q_{\boldsymbol{\phi}}(\boldsymbol{z} | \mathbb{X}_k) || p(\boldsymbol{z})] \right] + \mathcal{H}(q_{\boldsymbol{\phi}}(\boldsymbol{\xi} | \mathbb{X}_k)) \right\} + C, \tag{7}$$

where $\mathcal{H}$ is the entropy function, $\boldsymbol{\theta}$ and $\boldsymbol{\phi}$ are the learnable weights of the neural networks parameterizing generative and variational distributions. □

It is noteworthy that the entropy term $\mathcal{H}(q_{\boldsymbol{\phi}}(\boldsymbol{\xi} | \mathbb{X}_k))$ decreases during optimization of the ELBO, as it is not optimized in isolation. Therefore, CoDE-VAE learns uneven parameters $\pi_k$ as shown in Fig. 5.

---

[6]Alternatively, the lower bound could contain the $KL[q(\boldsymbol{\xi} | \mathbb{X}_k) || p(\boldsymbol{\xi})]$ divergence term. However, minimizing this divergence can tilt the parameters $\pi_k$ towards $1/K$. Instead, we let the data *speak* and optimize the entropy of the posterior distribution to learn $\boldsymbol{\pi}$.

## B. CoDE - A simple example

At this point it is important to note that all $\boldsymbol{\Sigma}^d$ in $\boldsymbol{\Sigma}_k$ are guaranteed to be full rank by construction, as $\sigma_{i,j} > 0$ for all $i, j$, where $\sigma_{i,i} = \sigma_i^2$. To see this, we need to show that the quadratic form $\boldsymbol{\beta}^T \boldsymbol{\Sigma}^d \boldsymbol{\beta} = 0$, is only satisfied for a zero-vector $\boldsymbol{\beta}$. Further, let $\kappa$ be the smallest $\sigma_{i,j}$ value, which is positive by construction. Therefore, $\sum_i \sum_j \beta_i \sigma_{i,j} \beta_j > \kappa \sum_i \sum_j \beta_i \beta_j$. Since $\kappa > 0$, the only solution that satisfies $\kappa \sum_i \sum_j \beta_i \beta_j = 0$ is the zero-vector $\boldsymbol{\beta}$. Since $\boldsymbol{\beta}^T \boldsymbol{\Sigma}^d \boldsymbol{\beta} = 0$ only for the zero-vector, $\boldsymbol{\Sigma}^d$ is positive definite and therefore invertible, and so is $\boldsymbol{\Sigma}_k$. $\boldsymbol{\Sigma}^d$ is still invertible even if its off-diagonal elements are 0 given that $\sigma_i^2 > 0$ for all i. Without loss of generality, assume that we observe bimodal data $\mathbb{X} = (x_1, x_2)$ with unimodal Gaussian data modalities. Therefore, the powerset is $\mathcal{P}(\mathbb{X}) = \{(x_1), (x_2), (x_1, x_2)\}$, implying that there are two expert distributions $q(z|x_1)$ and $q(z|x_2)$, and one unknown consensus distribution $q(z|x_1, x_2)$, i.e., $K = 2^M - 1$ distributions in total, where $M = 2$ is the number of modalities. Note that there are exactly $K$ scenarios where a set of modalities may not be available at test time. Therefore, we are interested in learning all $K$ distributions, so that at test time we can draw $\boldsymbol{z} \sim q(\boldsymbol{z}|\mathbb{X}_k)$, for $k = 1, \cdots, 3$, regardless if there are unavailable sets. In what follows, we drop the $k$ subscript and superscript in the formulae. Furthermore, assume that the unknown parameter is $\theta = 8$, the expert estimates and their uncertainty are $\mu_1 = 4$, $\mu_2 = 8$, $\sigma_1^2 = 3$, and $\sigma_2^2 = 1$, and that the estimates have correlation $\rho = 0.6$. Therefore, we have that

$$\boldsymbol{u} = \begin{bmatrix} 1 \\ 1 \end{bmatrix}, \quad \boldsymbol{\mu} = \begin{bmatrix} 4 \\ 8 \end{bmatrix} = \begin{bmatrix} \mu_1^1 \\ \mu_2^1 \end{bmatrix}, \quad \boldsymbol{e} = \begin{bmatrix} 4 - \theta \\ 8 - \theta \end{bmatrix} = \begin{bmatrix} e_1^1 \\ e_2^1 \end{bmatrix} \quad \text{and} \quad \boldsymbol{\Sigma} = \begin{bmatrix} 3 & 0.6 \cdot \sqrt{3} \cdot \sqrt{1} \\ 0.6 \cdot \sqrt{3} \cdot \sqrt{1} & 1 \end{bmatrix}.$$

Then we can calculate the consensus distribution using Lemma 2 as follows:

$$\mathcal{A} = \begin{bmatrix} 1 & 1 \end{bmatrix} \begin{bmatrix} \alpha_{1,1} & \alpha_{1,2} \\ \alpha_{2,1} & \alpha_{2,2} \end{bmatrix} \begin{bmatrix} 1 \\ 1 \end{bmatrix} = \alpha_{1,1} + \alpha_{2,1} + \alpha_{1,2} + \alpha_{2,2},$$

$$\mathcal{B} = \begin{bmatrix} 1 & 1 \end{bmatrix} \begin{bmatrix} \alpha_{1,1} & \alpha_{1,2} \\ \alpha_{2,1} & \alpha_{2,2} \end{bmatrix} \begin{bmatrix} \mu_1 \\ \mu_2 \end{bmatrix} = (\alpha_{1,1} + \alpha_{2,1})\mu_1 + (\alpha_{1,2} + \alpha_{2,2})\mu_2,$$

and $q(z|x_1, x_2) \sim \mathcal{N}\left( \frac{(\alpha_{1,1}+\alpha_{2,1})\mu_1 + (\alpha_{1,2}+\alpha_{2,2})\mu_2}{\alpha_{1,1}+\alpha_{1,2}+\alpha_{2,1}+\alpha_{2,2}}, \frac{1}{\alpha_{1,1}+\alpha_{1,2}+\alpha_{2,1}+\alpha_{2,2}} \right)$, where $\boldsymbol{\Sigma}^{-1} = \alpha_{i,j}$.

**CoDE subsumes PoE:** Now, let us assume that $\rho = 0$. Hence,

$$\boldsymbol{\Sigma} = \begin{bmatrix} \sigma_1^2 & 0 \\ 0 & \sigma_2^2 \end{bmatrix}, \quad \boldsymbol{\Sigma}^{-1} = \frac{1}{\sigma_1^2 \sigma_2^2} \begin{bmatrix} \sigma_2^2 & 0 \\ 0 & \sigma_1^2 \end{bmatrix} = \begin{bmatrix} 1/\sigma_1^2 & 0 \\ 0 & 1/\sigma_2^2 \end{bmatrix},$$

$\mathcal{A} = \tau_{1,1} + \tau_{2,2}$, $\mathcal{B} = \tau_{1,1}\mu_1 + \tau_{2,2}\mu_2$, and the consensus distribution is $q(z|x_1, x_2) \sim \mathcal{N}\left( \frac{\tau_{1,1}\mu_1 + \tau_{2,2}\mu_2}{\tau_{1,1}+\tau_{2,2}}, \frac{1}{\tau_{1,1}+\tau_{2,2}} \right)$, where $\tau_i = 1/\sigma_i^2$, like in (Wu & Goodman, 2018). Therefore, for $\rho = 0$ the CoDE parameters are simply PoE parameters.

As shown by (Winkler, 1981), we can write the mean of the consensus distribution as

$$\mu_{CoDE} = \omega_1 \mu_1 + \omega_2 \mu_2,$$

where $\omega_1 = \frac{(\sigma_2^2 - \rho\sigma_1\sigma_2)}{\sigma_1^2 + \sigma_2^2 - 2\rho\sigma_1\sigma_2}$ and $\omega_2 = \frac{(\sigma_1^2 - \rho\sigma_1\sigma_2)}{\sigma_1^2 + \sigma_2^2 - 2\rho\sigma_1\sigma_2}$, to show that the posterior parameter $\mu_{CoDE}$ is a weighted average of the expert parameters $\mu_1$ and $\mu_2$. Note that the weights can be negative depending on the value of $\rho$ with respect to $\sigma_1$ and $\sigma_2$. In the above example, for $\rho = 0.6$, the posterior mean is $\mu_{CoDE} = -0.02 * 4 + 1.02 * 8$ and for $\rho = 0$ the mean becomes $\mu_{CoDE} = 0.25 * 4 + 0.75 * 8$. In both cases, the weights sum up to 1. However, when the correlation between expert estimates is taken into account and the correlation is relatively high, the posterior mean leans even more towards the more accurate (with less variance) estimate. Fig. 7 shows both weights as a function of $\rho$ and, only for $\rho$ greater than 0.5, the weight for the less accurate estimate becomes negative. On the other hand, for $\rho = 0$ the weights are always positive and $\mu_{CoDE}$ is a convex function of the expert mean parameters. Similarly, we can write the variance of the consensus distribution as

$$\sigma_{CoDE}^2 = \frac{(1 - \rho^2)\sigma_1^2 \sigma_2^2}{\sigma_1^2 + \sigma_2^2 - 2\rho\sigma_1\sigma_2}.$$

Note that when the dependence between experts is neglected, i.e. $\rho = 0$, the variance of the consensus distribution is higher than it should be, as the denominator increases.

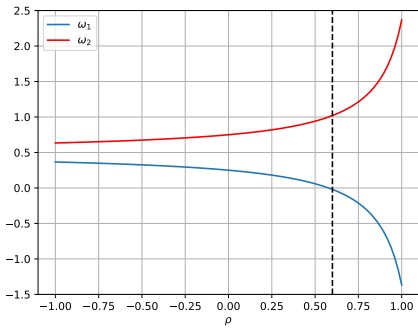

*Figure 7.* The consensus parameter $\mu_{CoDE}$ can be estimated as $\mu_{CoDE} = \omega_1 \mu_1 + \omega_2 \mu_2$, where the weights $\omega_1$ and $\omega_2$ are functions of the correlation parameter $\rho$.

**Two expert distributions and 2D joint representation $z$:**   In this case, $\boldsymbol{\theta} = (\boldsymbol{\theta}^1, \boldsymbol{\theta}^2)^T$ and the matrices for the consensus distributions are

$$
\boldsymbol{u} = \begin{bmatrix} 1 & 0 \\ 1 & 0 \\ 0 & 1 \\ 0 & 1 \end{bmatrix}, \quad
\boldsymbol{\mu} = \begin{bmatrix} \mu_1^1 \\ \mu_2^1 \\ \mu_1^2 \\ \mu_2^2 \end{bmatrix} = \begin{bmatrix} \boldsymbol{\mu}^1 \\ \boldsymbol{\mu}^2 \end{bmatrix}, \quad
\boldsymbol{e} = \begin{bmatrix} \mu_1^1 - \theta_1 \\ \mu_2^1 - \theta_1 \\ \mu_1^2 - \theta_2 \\ \mu_2^2 - \theta_2 \end{bmatrix} = \begin{bmatrix} \boldsymbol{e}_1 \\ \boldsymbol{e}_2 \end{bmatrix}, \quad
\boldsymbol{\Sigma} = \begin{bmatrix} \boldsymbol{\Sigma}^1 & \boldsymbol{0} \\ \boldsymbol{0} & \boldsymbol{\Sigma}^2 \end{bmatrix},
$$

where

$$
\boldsymbol{\Sigma}^d = \begin{bmatrix} \sigma_1^2 & \rho\sigma_1\sigma_2 \\ \rho\sigma_2\sigma_1 & \sigma_2^2 \end{bmatrix}
$$

for $d = 1, 2$ is the covariance matrix of the $d$-th dimension, and $\boldsymbol{0}$ is a 2x2 zero matrix. Therefore,

$$
\mathcal{A} = \boldsymbol{u}^T \boldsymbol{\Sigma}^{-1} \boldsymbol{u} = \begin{bmatrix} 1 & 1 & 0 & 0 \\ 0 & 0 & 1 & 1 \end{bmatrix} \begin{bmatrix} \alpha_{1,1} & \alpha_{1,2} & 0 & 0 \\ \alpha_{2,1} & \alpha_{2,2} & 0 & 0 \\ 0 & 0 & \alpha_{1,1} & \alpha_{1,2} \\ 0 & 0 & \alpha_{2,1} & \alpha_{2,2} \end{bmatrix} \begin{bmatrix} 1 & 0 \\ 1 & 0 \\ 0 & 1 \\ 0 & 1 \end{bmatrix}
$$

and

$$
\mathcal{B} = \boldsymbol{u}^T \boldsymbol{\Sigma}^{-1} \boldsymbol{\mu} = \begin{bmatrix} 1 & 1 & 0 & 0 \\ 0 & 0 & 1 & 1 \end{bmatrix} \begin{bmatrix} \alpha_{1,1} & \alpha_{1,2} & 0 & 0 \\ \alpha_{2,1} & \alpha_{2,2} & 0 & 0 \\ 0 & 0 & \alpha_{1,1} & \alpha_{1,2} \\ 0 & 0 & \alpha_{2,1} & \alpha_{2,2} \end{bmatrix} \begin{bmatrix} \mu_1^1 \\ \mu_2^1 \\ \mu_1^2 \\ \mu_2^2 \end{bmatrix}
$$

where $\boldsymbol{\Sigma}^{-1} = \alpha_{i,j}$ for $d = 1, 2$.

## C. Limitations

The dependence between expert distributions is captured in the $\rho$ parameter, which is found by cross-validation. For large and complex data, cross-validating $\rho$ can be computationally costly. Moreover, although CoDE is a principled Bayesian method, in which it is possible to choose different prior distributions and likelihood functions, it is not clear how multimodal VAEs can be trained with different choices than the ones we make for these. Finally, although CoDE-VAE has a relatively high computational cost $\mathcal{O}(2^M - 1)$, model training is feasible even for 5-modality datasets on a single GPU.

**Overhead added by CoDE:**   To calculate consensus distributions with the CoDE method, we only need to find the inverse of each $\boldsymbol{\Sigma}^d$ matrix, which is an affordable computation. The average processing time for one batch of the MNIST-SVHN-Text data using CoDE is 46 milliseconds, which is not significantly higher than using PoE (Wu & Goodman, 2018) (36 milliseconds). For PolyMNIST data, where $\boldsymbol{z} \in \mathbb{R}^{512}$, the average processing time for one batch is 1050 and 790 milliseconds using CoDE and PoE, respectively. Models are trained on a single A100 GPU.

---

**Algorithm 1** Minibatch version of the CoDE-VAE algorithm.

---

$\boldsymbol{\theta}, \boldsymbol{\phi}, \boldsymbol{\pi} \leftarrow$ initialize network parameters randomly and $\pi_k = 1/(2^M - 1)$. Define $\rho$ by cross-validation.
**repeat**
    $\mathbb{X}^i \leftarrow$ Random minibatch with all data modalities
    $\boldsymbol{\epsilon} \leftarrow$ Random samples from $\mathcal{N}(\mathbf{0}, \mathbf{1})$
    **for** $\mathbb{X}_k^i \in \mathcal{P}(\mathbb{X}^i)$ **do**
        $\boldsymbol{\Sigma}_k \leftarrow$ Define $\boldsymbol{\Sigma}^d$ using the encoder outputs $\sigma_i^2$ and $\rho\sigma_i\sigma_j$, for $d = 1, \cdots, D$.
        $\mathcal{A}_k, \mathcal{B}_k \leftarrow$ Use encoder outputs $\mu_i$ and find the inverse of $\boldsymbol{\Sigma}_k$
        $q(\boldsymbol{z}|\mathbb{X}_k^i) \leftarrow$ Expert and consensus distributions         ▷Apply Lemma 2
        $\nabla\mathcal{L}_k(\boldsymbol{\theta}, \boldsymbol{\phi}, \pi_k, \mathbb{X}^i, \boldsymbol{\epsilon}) \leftarrow$ Get gradients         ▷Apply Lemma 3
    **end for**
    $\boldsymbol{\theta}, \boldsymbol{\phi}, \boldsymbol{\pi} \leftarrow$ Update parameters with the Adam optimizer
**until** Convergence of $\boldsymbol{\theta}, \boldsymbol{\phi}, \boldsymbol{\pi}$
**return** $\boldsymbol{\theta}, \boldsymbol{\phi}, \boldsymbol{\pi}$

---

## D. Experiments

### D.1. COMMON MODEL TRAINING

We optimize Eq. 3 by stochastic gradient descent using the reparameterization trick (Kingma & Welling, 2014) for estimating gradients (see Algorithm 1), and the Adam optimizer (Kingma & Ba, 2017) is used on all datasets. Models are trained on single A100 GPUs with AMD EPYC Milan processors with 24 cores. We find the optimal correlation parameter $\rho$ by cross-validation using the values $[0, 0.2, 0.4, 0.6, 0.8]$ and the off-diagonal in the covariance matrices $\boldsymbol{\Sigma}^d$ in Lemma 1 are specified as $\sigma_{j,i} = \rho\sigma_j\sigma_i$. We cross-validate only positive $\rho$ values, as it is reasonable to infer that the dependency between experts arises from common information about their underlying object. Following previous research, we scale the Kullback-Leibler divergence terms in Eq. 3 by a regularization coefficient $\beta$ (Higgins et al., 2016), i.e. $\beta KL[q_\phi(\boldsymbol{z}|\mathbb{X}_k)||p(\boldsymbol{z})]$, as it has a significant impact on the evaluation of the model. The $\beta$ value is found by cross-validation using the values $[0.1, 1, 5, 10, 15, 20]$. We also consider $\beta = 2.5$ in the PolyMNIST data to replicate the setting in (Palumbo et al., 2023). For all datasets, we assume that the prior distribution is an isotropic Gaussian distribution, and the expert distributions are assumed to be multivariate Gaussian distributions with diagonal covariance matrix. All consensus distributions are approximated using Lemma 2. Finally, all experiments report the average performance and standard deviations of 3 different runs and for the benchmark models we use the following original implementation codes: MMVAE https://github.com/iffsid/mmvae/tree/public; MVAE, mmJSD, and MoPoE https://github.com/thomassutter/MoPoE; MVT-CAE https://github.com/gr8joo/MVTCAE/tree/master; MMVAE+ https://github.com/epalu/mmvaeplus/tree/new_release.

### D.2. EVALUATION CRITERIA

**Linear classification:** We use the `LogisticRegression` class in `sklearn` to fit linear classifiers with 500 latent representations of all subsets $\mathbb{X} \in \mathcal{P}(\mathbb{X})$. The only parameters that we specify are `solver='lbfgs'`, `multi_class='auto'` and `max_iter=3000`. To test the performance of the classifiers on the entire test set, we use `accuracy_score`, which is available in `sklearn`, for the MNIST-SVHN-Text and PolyMNIST data. See the released code for further details.

**Coherence:** For each of the modalities in the datasets, we train classifier networks that have the same architecture as their encoder, using original observations of the same modality. Then we generate modalities conditioned on latent representations of each subset $\mathbb{X}_k \in \mathcal{P}(\mathbb{X})$, including the empty set, in which case $\boldsymbol{z} \sim p(\boldsymbol{z})$. To evaluate unconditional coherence, we classify the generated modalities conditioned on representations of the prior, calculate the number of generated modalities classified as having the same label, and divide it by the number of total modalities generated. To measure conditional coherence, we classify generated modalities conditioned on representations of subsets where the modality being classified is not present, e.g., MNIST images can only be generated conditioned on subsets containing SVHN, Text, and SVHN&Text modalities, and calculate the `accuracy_score` for the MNIST-SVHN-Text and PolyMNIST data.

**Fréchet inception distance (FID):** We use a TensorFlow pre-trained inception network to calculate the FID scores. Note that we tested our TensorFlow implementation with that of (Daunhawer et al., 2022), which is coded in PyTorch, and for

*Table 3.* MNIST encoder and decoder layers. All layers are linear with ReLU activations. Finally, the number of input and output dimensions in each layer is shown in the columns #F.In and #F.Out, respectively.

| | Encoder | | | | Decoder | | |
|---|---|---|---|---|---|---|---|
| Layer | Type | #F.In | #F.Out | Layer | Type | #F.In | #F.Out |
| 1 | linear | 784 | 400 | 1 | linear | 20 | 400 |
| 2a | linear | 400 | 20 | 2 | linear | 400 | 784 |
| 2b | linear | 400 | 20 | | | | |

*Table 4.* SVHN encoder and decoder layers. The last column for each model specifies the kernel size, stride, padding, and dilation. All layers are 2D convolutional (conv) and upconvolutional (upconv) in the encoder and decoder, respectively, with ReLU activations. Finally, the number of input and output dimensions in each layer is shown in the columns #F.In and #F.Out, respectively.

| | Encoder | | | | | Decoder | | | |
|---|---|---|---|---|---|---|---|---|---|
| Layer | Type | #F.In | #F.Out | Spec. | Layer | Type | #F.In | #F.Out | Spec. |
| 1 | conv | 3 | 32 | (4,2,1,1) | 1 | linear | 20 | 128 | |
| 2 | conv | 32 | 64 | (4,2,1,1) | 2 | upconv | 128 | 64 | (4,2,0,1) |
| 3 | conv | 64 | 64 | (4,2,1,1) | 3 | upconv | 64 | 64 | (4,2,1,1) |
| 4 | conv | 64 | 128 | (4,2,0,1) | 4 | upconv | 64 | 32 | (4,2,1,1) |
| 5a | linear | 128 | 20 | | 5 | upconv | 32 | 3 | (4,2,1,1) |
| 5b | linear | 128 | 20 | | | | | | |

a sample of PolyMNIST images, we obtained the same values. See the released code for details of the implementation. The conditional FID scores are calculated using generated modalities conditioned on representations of all subsets, while the unconditional FID scores are calculated using generated modalities conditioned on representations from the prior distribution. In both cases, generated images are compared with original ones. The conditional FID scores reported in Table 1 and in Fig. 4 are averages of generated images conditioned on all subsets and all modalities, while the conditional FID scores in Fig. 3 for modality $m_0$ are averages of generated images conditioned on all subsets. Finally, the scores in Table 2 correspond to generated images conditioned on the subset containing the caption modality.

### D.3. MNIST-SVHN-TEXT

**Data and training details:** The dataset consists of handwritten digits, images, and a text strings corresponding to an underlying digit. The triples are created in a many-to-many mapping, therefore, there are 1,121,360 and 200,000 observations in the train and test sets, respectively. The digit text has 8 characters and its initial position is chosen randomly, leaving the remaining characters blank. We train our CoDE-VAE model with the Adam optimizer with default values and a learning rate of 0.001, using mixed-precision to speed up model training. Both image modalities are assumed to have Laplace likelihoods, whereas the text modality is assumed to have a categorical likelihood. The dimension of the latent space is set to 20, as in (Sutter et al., 2020; 2021; Palumbo et al., 2023; Mancisidor et al., 2024). For a fair comparison of all models, we follow a similar approach as (Palumbo et al., 2023) to select the dimension of the modality-specific latent space in MMVAE+. That is, divide the total number of dimensions in the latent space assumed by models without modality-specific variables by the number of modalities.

We use weights to scale the decoder networks, as the distributions for the modalities have different scales. The weights are calculated as in (Sutter et al., 2021), where the modality with the highest dimension is set to 1 and all the others are scaled by their relative ratio to the dimension of the data. Hence, the weights we use are 1, 3.9, and 384 for the SVHN, MNIST, and text decoders, respectively. Similarly, we scale the entropy loss by 1,000, a value which we found by cross-validation and that works well for all datasets in this research. The architectures that we use in the encoder and decoder networks are shown in Tables 3-5.

**Classification and qualitative results:** Fig. 8 shows the classification accuracy of a linear classifier trained with latent representations for all subsets $\mathbb{X}_k \in \mathcal{P}(\mathbb{X})$, averaged over equal cardinality subsets. We omit standard deviations as they are smaller than 0.01 for all models. Note that only the models using PoE and CoDE are able to achive higher classification accuracy as the number of expert increases. On the other hand, models using MoE achieves a flat performance despite having more modalities to approximate the joint posterior distributions. The CoDE-VAE model achieves slightly higher

*Table 5.* Text encoder and decoder layers. The last column for each model specifies the kernel size, stride, padding, and dilation. All layers are 1D convolutional (conv) and upconvolutional (upconv) in the encoder and decoder, respectively, with ReLU activations. Finally, the number of input and output dimensions in each layer is shown in the columns #F.In and #F.Out, respectively.

| | Encoder | | | | | Decoder | | | |
|---|---|---|---|---|---|---|---|---|---|
| Layer | Type | #F.In | #F.Out | Spec. | Layer | Type | #F.In | #F.Out | Spec. |
| 1 | conv | 71 | 128 | (1,1,0,1) | 1 | linear | 20 | 128 | |
| 2 | conv | 128 | 128 | (4,2,1,1) | 2 | upconv | 128 | 128 | (4,1,0,1) |
| 3 | conv | 128 | 128 | (4,2,0,1) | 3 | upconv | 128 | 128 | (4,2,1,1) |
| 4a | linear | 128 | 20 | | 4 | conv | 128 | 71 | (1,1,0,1) |
| 4b | linear | 128 | 20 | | | | | | |

classification accuracy in subsets with 2 and 3 modalities compared to that of all benchmark models. Finally, Fig. 9 shows samples of modalities that are conditionally generated on representations of the prior distribution (top row), and on representations of the consensus distribution $q(z|\overline{\overline{\mathbb{X}}} = 3)$ (bottom row).

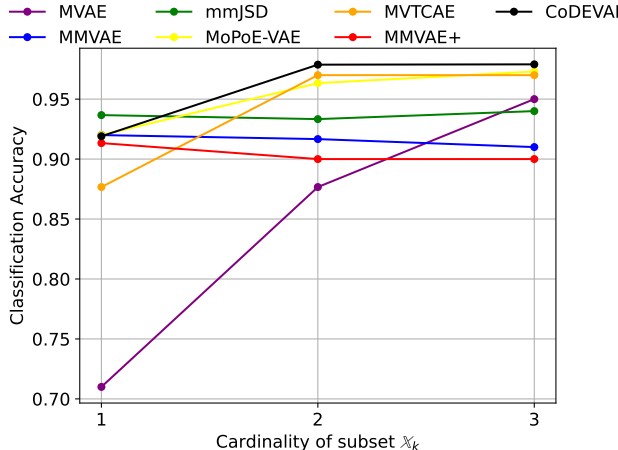

*Figure 8.* Classification accuracy of a linear classifier trained with latent representations for all subsets $\mathbb{X}_k \in \mathcal{P}(\mathbb{X})$, averaged over equal cardinality subsets.

*Table 6.* Optimal parameters for all models in the classification on MNIST-SVHN-Text data.

| | MVAE | MMVAE | mmJSD | MoPoE-VAE | MVTCAE | MMVAE+ | CoDE-VAE |
|---|---|---|---|---|---|---|---|
| $\beta$ | 5 | 0.1 | 5 | 10 | 20 | 5 | 5 |
| $\rho$ | | | | | | | 0.4 |

### D.4. POLYMNIST

**Data and training details:** the PolyMNIST dataset is built upon MNIST by varying the original background. A random 28x28 crop from 5 different images is used as the background to form a 5-modality dataset. Links to the images are provided in (Sutter et al., 2021), where the dataset was first introduced. Note that we keep the same order in the links of the images to define modality $m_0$, $m_1$, etc. Training and test sets have 60,000 and 10,000 images, respectively. We use the Adam optimizer with default values and a learning rate of 0.001 to train CoDE-VAE models using mixed-precision. Since all modalities have the same dimension, it is not necessary to scale the decoders in this case, but the entropy term is again scaled by 1,000. The architectures of the encoder and decoder are shown in Table 8, which are similar to the architectures in (Sutter et al., 2021) and not in (Daunhawer et al., 2022). We assume Laplace likelihoods. The dimensionality of the latent representations is set to 512 as in (Sutter et al., 2021; Daunhawer et al., 2022; Palumbo et al., 2023).

The results in Fig. 4 are based on all 5 modalities of the PolyMNIST data, while the results in Fig. 3 and in Fig. 6 show

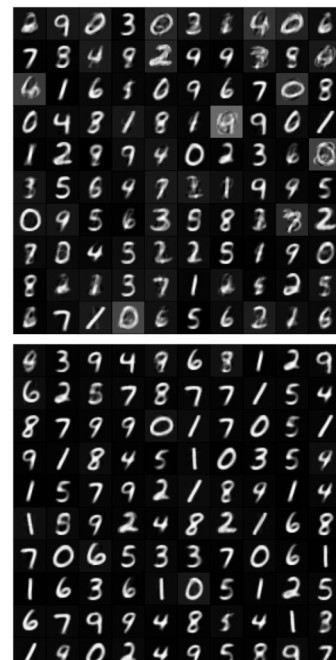
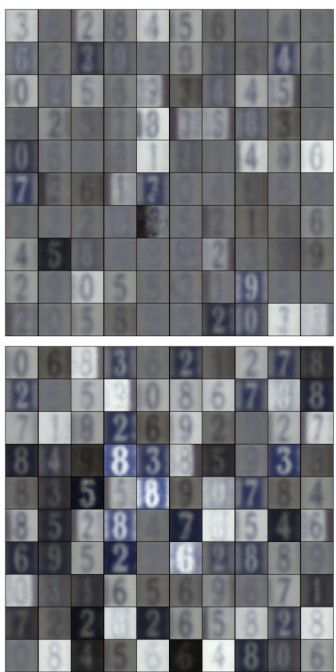
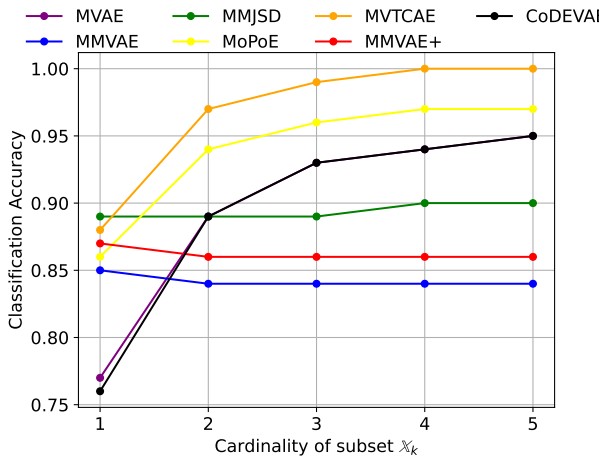

*Figure 9.* Random generation of digits corresponding to the MNIST, SVHN, and Text modalities. The decoders are conditioned on representations from the prior (top row) and the consensus $q(\boldsymbol{z}|\overline{\overline{\mathbb{X}}} = 3)$ distribution.

*Figure 10.* Classification accuracy of a linear classifier trained with latent representations for all subsets $\mathbb{X}_k \in \mathcal{P}(\mathbb{X})$, averaged over equal cardinality subsets.

*Table 7.* Optimal parameters for all models in the classification on PolyMNIST data.

|  | MVAE | MMVAE | mmJSD | MoPoE-VAE | MVTCAE | MMVAE+ | CoDE-VAE |
|---|---|---|---|---|---|---|---|
| $\beta$ | 1 | 1 | 1 | 1 | 5 | 1 | 5 |
| $\rho$ |  |  |  |  |  |  | 0.4 |

FID scores obtained by the CoDE-VAE model trained with 2, 3, 4, and 5 modalities, i.e. we train our proposed model with four *versions* of the data depending on the number of modalities considered, which we ordered as follows:

- 2 modalities $\rightarrow \{m_0, m_1\}$

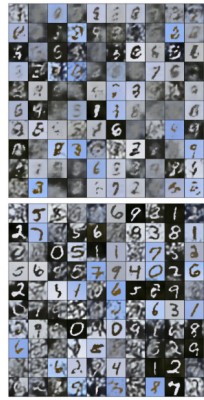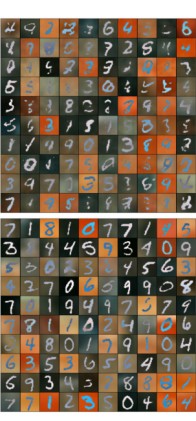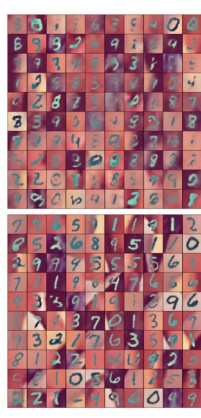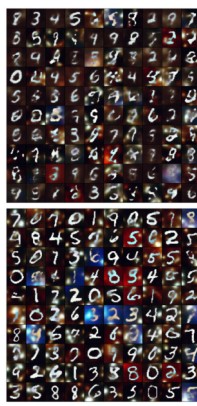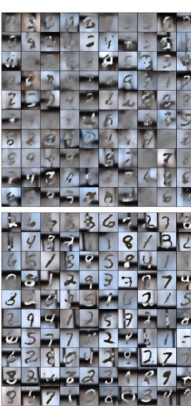

*Figure 11.* Random generation of digits for all modalities. The decoders are conditioned on representations from the prior (top row) and the consensus distribution $q(z|\overline{\overline{\mathbb{X}}} = 5)$.

- 3 modalities $\rightarrow \{m_0, m_1, m_2\}$

- 4 modalities $\rightarrow \{m_0, m_1, m_2, m_3\}$

- 5 modalities $\rightarrow \{m_0, m_1, m_2, m_3, m_4\}$

Given that we cross-validate $\beta$, $\rho$, and $\pi$ (for the ablation experiments) we trained 120 different configurations assuming 2, 3, 4, and 5 modalities, i.e. 480 experiments that we run 3 different times each. Following a similar approach to (Daunhawer et al., 2022), we compared the quality of the generative model choosing the architecture with the smallest FID scores.

**Classification and qualitative results:** Classification results are shown in Fig. 10 where standard deviations are omitted as they are smaller than 0.01 for all models. We observe a similar trend as in the classification results for the MNIST-SVHN-Text data. Models using MoE are not able to learn extra information that is useful for classification as the number of modalities increases. Note that our results for the mmJSD model do not agree with those presented in (Huang et al., 2022) that show increasing accuracy similar to that of MVTCAE. Such a trend is not common in models using MoE and, therefore, we believe that they are wrong. The MVTCAE model achieves a perfect classification accuracy for subsets with 4 and 5 models, followed by MoPoE, CoDE-VAE and MVAE.

Fig. 11 shows samples of all 5 modalities that are conditionally generated on representations of the prior distribution (top row), and on representations of the consensus distribution $q(z|\overline{\overline{\mathbb{X}}} = 5)$ (bottom row). In Fig. 12, we show random generated digits corresponding to modality 1 (top row) and modality 2 (bottom row), as a function of the number of modalities in the dataset, i.e. the first column shows generated images with a CoDE-VAE model trained with 2 modalities, while the last column shows images generated with a model trained with 5 modalities. Finally, Fig. 13 shows the average coherence and classification accuracy as a function of correlation $\rho$, obtained with CoDE-VAE models trained with 2, 3, 4, and 5 modalities. The average is calculated over all subsets in each case, i.e. 3, 7, 15, and 31 subsets, respectively. In all cases, the coherence and classification is higher when the correlation between experts is taken into account, except when CoDE-VAE is trained with 5 modalities.

Finally, Fig. 14 shows random generations of the modality $m_0$ by unimodal VAE (top row) and CoDE-VAE, which were used to calculate the FID scores in Section 4.2 Generative quality gap. The generations of both models are visually indistinguishable.

D.5. CUB

**Data and training details:** the CUB data used in this research are composed of two modalities: bird images and 10 different detailed descriptions for each image. Therefore, there are in total 117,880 pairs of image-captions, where 88,550 are used for model training and 29,330 for testing. This bi-modal version of the CUB dataset has been used in (Shi et al., 2019; Daunhawer et al., 2022; Palumbo et al., 2023). However, (Shi et al., 2019) uses a simplified version that replaces bird images with ResNet embeddings. The architectures for the encoder and decoder caption modality are in Table 14, and

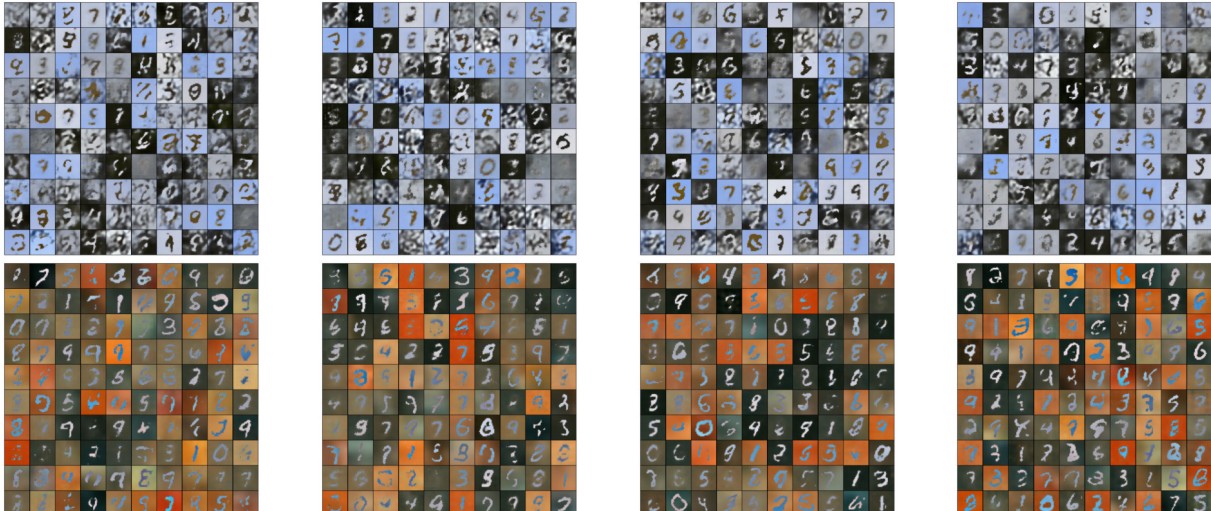

*Figure 12.* Random generation of digits in modality $m_0$ (top row) and modality $m_1$ (bottom row). The first column shows images of digits generated with a CoDE-VAE model trained with 2 modalities. Similarly, the second, third, and last columns show images generated by CoDE-VAE trained with 3, 4, and 5 modalities.

*Table 8.* PolyMNIST encoder and decoder layers. The last column for each model specifies the kernel size, stride, padding, and dilation. All layers are 2D convolutional (conv) and upconvolutional (upconv) in the encoder and decoder, respectively, with ReLU activations. Finally, the number of input and output dimensions in each layer is shown in the columns #F.In and #F.Out, respectively.

| Encoder | | | | | Decoder | | | | |
|---|---|---|---|---|---|---|---|---|---|
| Layer | Type | #F.In | #F.Out | Spec. | Layer | Type | #F.In | #F.Out | Spec. |
| 1 | conv | 3 | 32 | (3,2,1,1) | 1 | linear | 512 | 2048 | |
| 2 | conv | 32 | 64 | (3,2,1,1) | 2 | upconv | 2048 | 64 | (3,2,0,1) |
| 3 | conv | 64 | 128 | (3,2,1,1) | 3 | upconv | 64 | 32 | (3,2,1,1) |
| 4a | linear | 128 | 512 | | 4 | upconv | 32 | 3 | (3,2,1,1) |
| 4b | linear | 128 | 512 | | | | | | |

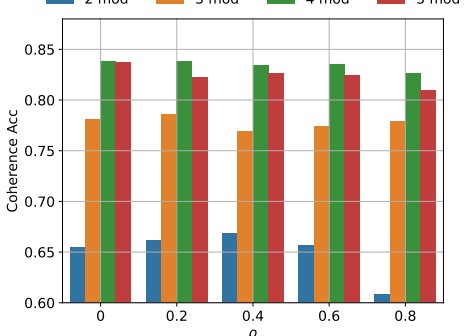

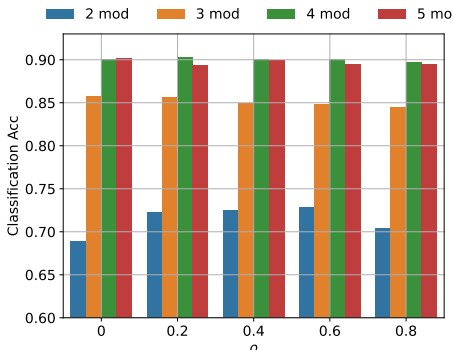

*Figure 13.* Average coherence and classification accuracy as a function of correlation $\rho$, obtained with CoDE-VAE models trained with 2, 3, 4, and 5 modalities. The average is calculated over all subsets in each case, i.e. 3, 7, 15, and 31 subsets, respectively.

for the image modality please see the released code. The dimensionality of the latent space is set to 64 as in (Daunhawer et al., 2022; Palumbo et al., 2023) and the weights to scale the decoders are 0.0026, and 1.0 for the image and caption modalities, respectively.

We follow the approach introduced in (Palumbo et al., 2023) to calculate the conditional coherence. First, we construct the following caption *"this bird is completely <color>"*, where color is any color in {*white, yellow, red, blue, green, gray,*

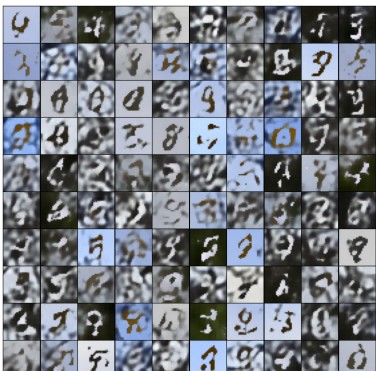

(a) Unconditional generation of modality $m_0$ by VAE.

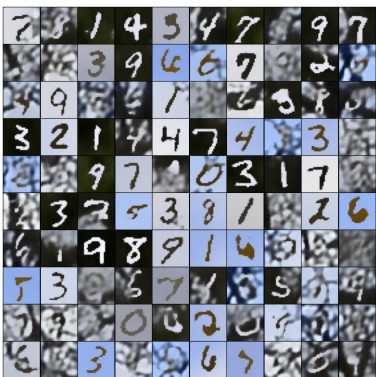

(b) Conditional generation of modality $m_0$ by VAE.

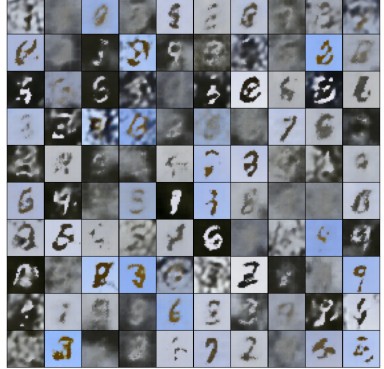

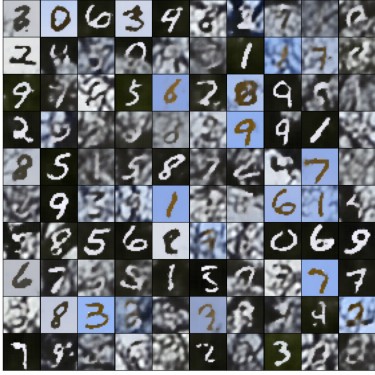

(c) Unconditional generation of modality $m_0$ by CoDE-VAE.   (d) Conditional generation of modality $m_0$ by CoDE-VAE.

*Figure 14.* The top row shows random generations by unimodal VAE, while the bottom row corresponds to generations by CoDE-VAE. The conditional generations by CoDE-VAE are based on the full subsets.

*brown, black*}. Then, for each of the eight captions, we generate ten images (eighty in total). Finally, we count the number of pixels that are within the boundaries of hue, saturation, and value (HSV) colors in Table 13, and an image is said to be coherent if any of the two classes of the highest pixel count is the same as the color in the initial caption. Note that we use the library `cv2` and the method `cvtColor` to obtain the HSV values of the generated images.

**Qualitative results:**   Fig. 15 shows the generated images for the coherent test previously explained. The row at the top shows the captions that are used to generate the image modality. On the otter hand, Fig. 16 shows images generated from original images in the CUB test sets, which are in the first row. Both grid of plots show that the generated images are able to capture the details in the modality that they are conditioned on. Finally, Fig. 17 shows generated images from original captions in the test set.

### E. Additional Experiments

Section 4 presents the main relevant benchmark models and the results against which the performance of these models should be compared. However, for completeness, this section analyses the performance of CoDE-VAE on the real and challenging bi-modal CelebA data (Sutter et al., 2020), and compares the generative quality and generative coherence of CoDE-VAE with that of the clustering multimodal VAE (CMVAE) model introduced in (Palumbo et al., 2024). Although the contribution of the CMVAE model is relevant to the field of multimodal VAEs, its line of research is orthogonal to CoDE-VAE, as CMVAE leverages clustering structures in the latent space by introducing a flexible prior distribution based on a mixture model and uses diffusion decoders.

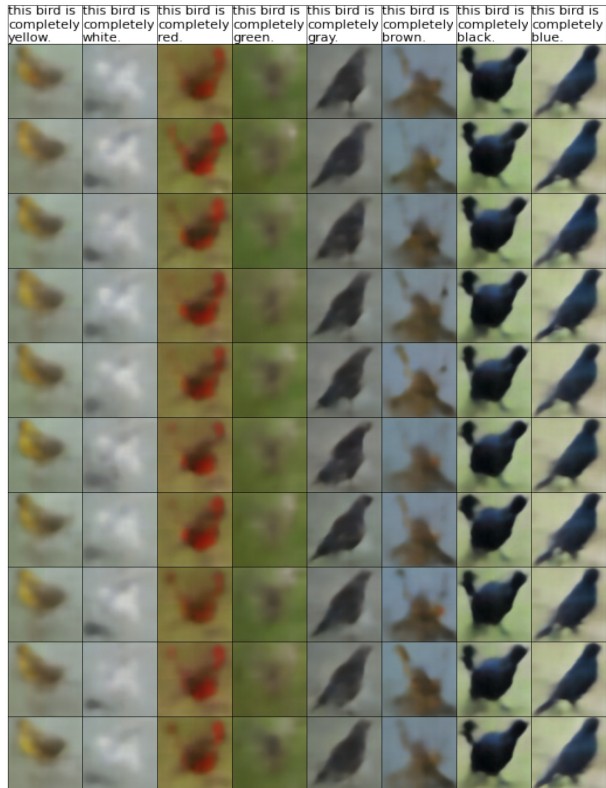

Figure 15. Caption-to-image generation used in the coherence test.

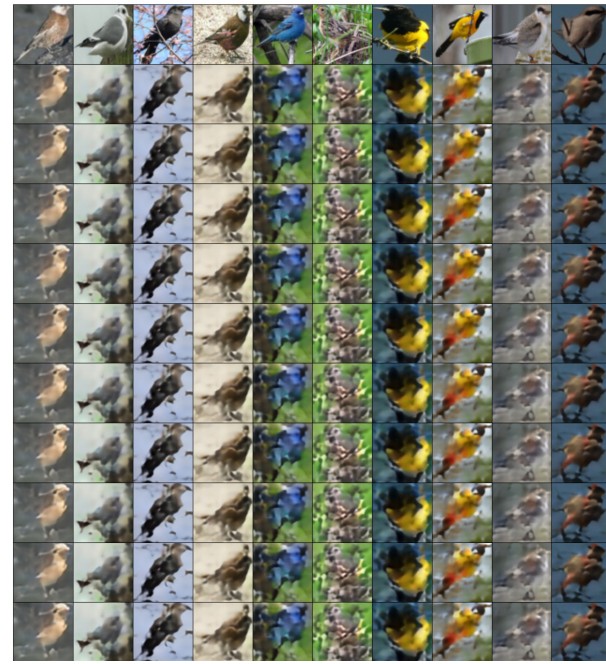

Figure 16. Image-to-image generation. Original images in the first row.

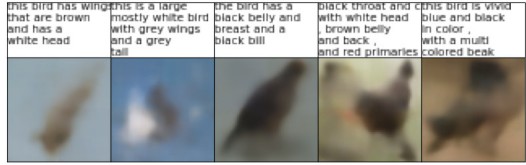

Figure 17. Caption-to-image generation for the captions in the first row.

### E.1. CELEBA

We test the performance of CoDe-VAE and MMVAE+ on the bi-modal CelebA data, which are composed of images and text descriptions for each image. The description modality is based on 40 attributes that describe each image and is fixed at 256 characters. For images with shorter descriptions, the remaining spaces are filled with a "*" character. The train and set sets have 162,560 and 19,712 observations, respectively. We train the CoDE-VAE model with default values in the Adam optimizer and a learning rate of 0.0001, using mixed precision and assuming a Laplace likelihood for the image modality and a categorical likelihood for the text modality. Furthermore, we assume $\rho = 0.6$, $\beta = 1$, and a latent space with 32 dimensions. For MMVAE+, we assume $\beta = 1$ and both modality-specific and common latent variables have 16 dimensions each. Hence, the decoders in both models generate modalities based on 32 dimensions. The architectures for the encoder and decoder are based on residual networks (He et al., 2016) and are shown in Tables 9 and 10.

Table 11 shows the performance of CoDe-VAE and MMVAE+ in terms of generative quality, generative coherence, and classification accuracy. CoDE-VAE achieves better generative quality and classification accuracy even when the $\rho$ parameter was not cross-validated. Fig. 18 shows some random faces, which are conditionally generated on representations of the prior distribution (left plot) and on representations of the consensus distribution $q(z|\bar{\bar{\mathbb{X}}} = 2)$ (right panel). Note that some of the images conditionally generated on the consensus distribution $q(z|\bar{\bar{\mathbb{X}}} = 2)$, have not only sharper faces, but also sharper backgrounds. On the other hand, the top row of Figure 19 shows faces conditionally generated on representations

*Table 9.* CelebA encoder and decoder layers for images. The last column for each model specifies the kernel size, stride, padding, and dilation. Layers are 2D convolutional (conv), upconvolutional (upconv), residual blocks (res), and upconvolutional residual blocks (res_upconv), with ReLU activations. Finally, the number of input and output dimensions in each layer is shown in the columns #F.In and #F.Out, respectively.

| | Encoder | | | | | Decoder | | | |
|---|---|---|---|---|---|---|---|---|---|
| Layer | Type | #F.In | #F.Out | Spec. | Layer | Type | #F.In | #F.Out | Spec. |
| 1 | conv | 3 | 128 | (3,2,1,1) | 1 | linear | 32 | 640 | |
| 2 | res | 128 | 256 | (4,2,1,1) | 2 | res_upconv | 640 | 512 | (4,1,0,1) |
| 3 | res | 256 | 384 | (4,2,1,1) | 3 | res_upconv | 512 | 384 | (4,1,1,1) |
| 4 | res | 384 | 512 | (4,2,1,1) | 4 | res_upconv | 384 | 256 | (4,1,1,1) |
| 5 | res | 512 | 640 | (4,2,1,1) | 5 | res_upconv | 256 | 128 | (4,1,1,1) |
| 6a | linear | 640 | 32 | | 6 | upconv | 128 | 3 | (3,2,1,1) |
| 6b | linear | 640 | 32 | | | | | | |

*Table 10.* CelebA encoder and decoder layers for text descriptions. The last column for each model specifies the kernel size, stride, padding, and dilation. Layers are 1D convolutional (conv), upconvolutional (upconv), residual blocks (res), and upconvolutional residual blocks (res_upconv), with ReLU activations. Finally, the number of input and output dimensions in each layer is shown in the columns #F.In and #F.Out, respectively.

| | Encoder | | | | | Decoder | | | |
|---|---|---|---|---|---|---|---|---|---|
| Layer | Type | #F.In | #F.Out | Spec. | Layer | Type | #F.In | #F.Out | Spec. |
| 1 | conv | 71 | 128 | (4,2,1,1) | 1 | linear | 32 | 640 | |
| 2 | res | 128 | 256 | (4,2,1,1) | 2 | res_upconv | 640 | 640 | (4,1,0,1) |
| 3 | res | 256 | 384 | (4,2,1,1) | 3 | res_upconv | 640 | 640 | (4,2,1,1) |
| 4 | res | 384 | 512 | (4,2,1,1) | 4 | res_upconv | 640 | 512 | (4,2,1,1) |
| 5 | res | 512 | 640 | (4,2,1,1) | 5 | res_upconv | 512 | 384 | (4,2,1,1) |
| 6 | res | 640 | 640 | (4,2,1,1) | 5 | res_upconv | 384 | 256 | (4,2,1,1) |
| 7 | res | 640 | 640 | (4,2,0,1) | 5 | res_upconv | 256 | 128 | (4,2,1,1) |
| 8a | linear | 640 | 32 | | 6 | upconv | 128 | 71 | (4,2,1,1) |
| 8b | linear | 640 | 32 | | | | | | |

*Table 11.* Model performance of CoDE-VAE and MMVAE+ on the CelebA data.

| | CoDE-VAE | MMVAE+ |
|---|---|---|
| Conditional FID ($\downarrow$) | **92.11** $\pm 0.61$ | 97.30 $\pm 0.40$ |
| Unconditional FID ($\downarrow$) | **87.41** $\pm 0.36$ | 96.91 $\pm 0.42$ |
| Conditional Coherence ($\uparrow$) | 0.38 $\pm 0.001$ | **0.46** $\pm 0.001$ |
| Unconditional Coherence ($\uparrow$) | 0.23 $\pm 0.003$ | **0.31** $\pm 0.030$ |
| Classification ($\uparrow$) | **0.38** $\pm 0.002$ | 0.37 $\pm 0.003$ |

of the expert distribution $q(\boldsymbol{z}|\mathbb{X}_{text})$. Note that some attributes, such as gender, smile, 5 o'clock shadow, are relatively easy to capture in the generated face. The bottom row of Figure 19 shows faces conditionally generated on representations of the consensus distribution $q(\boldsymbol{z}|\overline{\overline{\mathbb{X}}} = 2)$. For both cases, we added the text modality that describes the face attributes.

### E.2. POLYMNIST

We follow the same experimental setup as in Section 4.2 and compare the generative quality and generative coherence of all models, including CMVAE, which is shown in Fig. 20. CMVAE achieves significantly higher performance in unconditional coherence, as it uses a flexible prior distribution composed of a mixture model (one component for each cluster in the latent space). Such a prior distribution is certainly more informative compared to the non-informative isotropic Gaussian used in CoDE-VAE and most of benchmark models. However, CoDE-VAE achieves about the same performance in conditional coherence and conditional FID scores. Interestingly, despite its more complex architecture, CMVAE is not able to achieve significantly better unconditional and conditional FID scores.

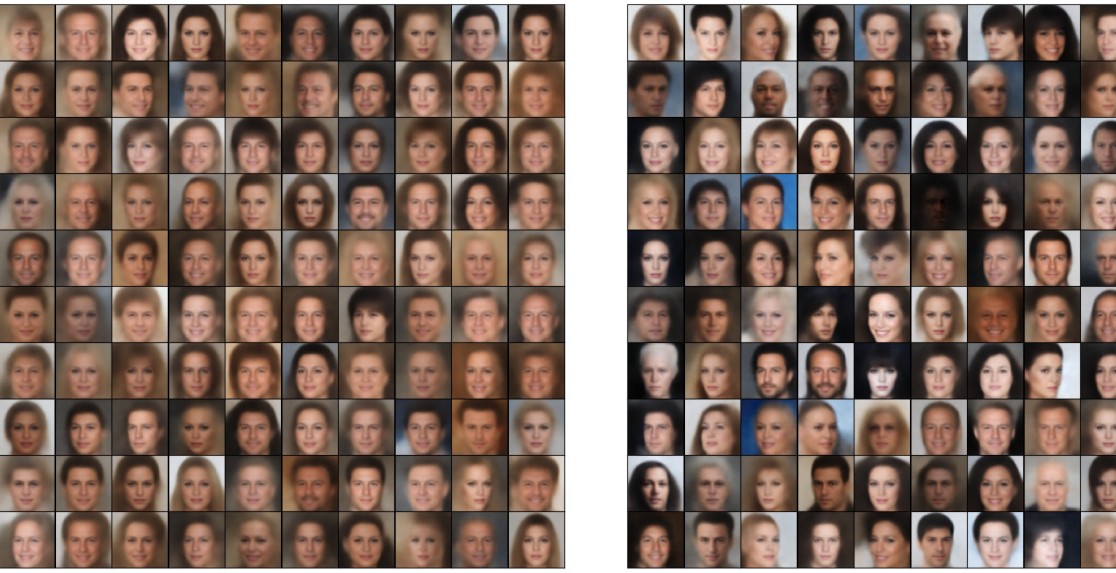

*Figure 18.* Random generation of faces. The decoder is conditioned on representations from the prior (left plot) and the consensus $q(z|\overline{\overline{\mathbb{X}}} = 2)$ distribution (right plot).

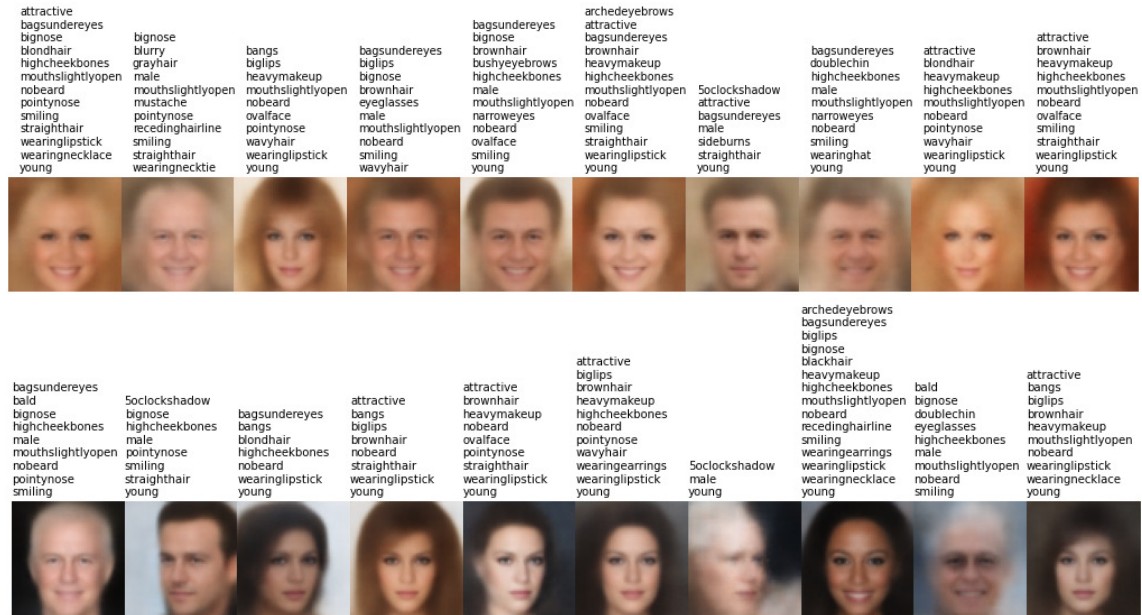

*Figure 19.* Conditionally generated faces on representations of the expert distribution $q(z|\mathbb{X}_{text})$ (top row) and the consensus distribution $q(z|\overline{\overline{\mathbb{X}}} = 2)$ (bottom row), where we added the text modality that describes the face attributes.

## F. Effect of weights $\pi$ and correlation $\rho$

We train the CoDE-VAE model using equal weights $\pi$ and assuming $\rho = 0$ (Baseline), and learning the weights $\pi$ and cross validating $\rho \in [0.2, 0.4, 0.6, 0.8]$ (Optimal), to assess its effect on unconditional coherence and log-likelihoods across different $\beta$ values. Fig. 21 shows that for all $\beta$ values, but $\beta = 1$, the Optimal models achieve higher coherence, higher log-likelihood, or both, relative to the Baseline models when using the PolyMNIST data. For $\beta = 1$, the coherence of the Baseline model is slightly higher, but the Optimal model has higher likelihood.

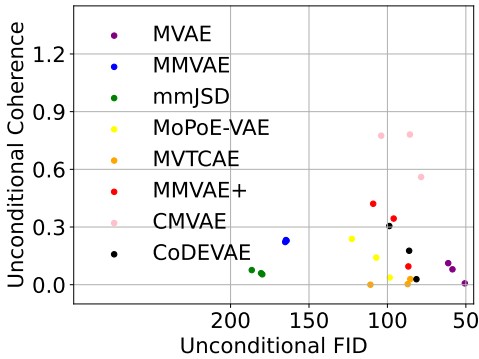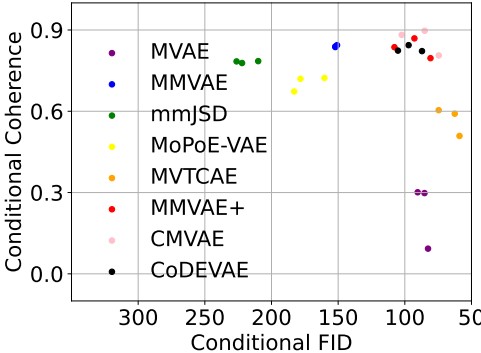

*Figure 20.* Trade-off between generative coherence (↑) and generative quality (↓) for $\beta \in [1, 2.5, 5]$ on the PolyMNIST dataset.

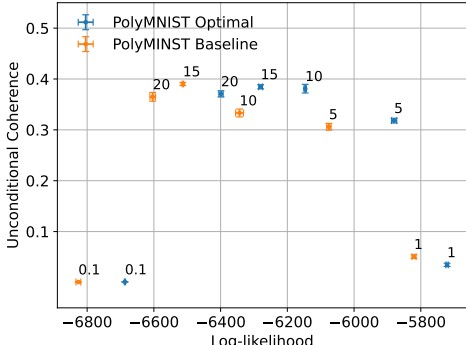

*Figure 21.* Effect of learning the $\pi_k$ weights and cross-validating $\rho \in [0.2, 0.4, 0.6, 0.8]$ across different $\beta$ values, which are annotated at the top-right of the scatters.

*Table 12.* Effect of edge case correlated modalities, assuming $\rho = 0$ and $\rho = 0.9$, on generative quality measured by FID scores (↓).

|              | 0%    | 25%   | 95%   |
| ------------ | ----- | ----- | ----- |
| $\rho = 0.0$ | 29.00 | 31.27 | 48.22 |
| $\rho = 0.9$ | 26.12 | 29.90 | 53.68 |

Table 15 shows the generative coherence, generative quality, and classification accuracy as a function of $\rho$ on the MNIST-SVHN-Text dataset. All values are calculated at $\beta = 20$. All metrics show a significant improvement for $\rho > 0$.

**Edge case correlated modalities:** To better understand the behavior of CoDE-VAE, we use the modality $m_1$ in PolyM-NIST in the following way. We apply three different levels of noise to the modality $m_1$: 0 %, 25%, and 95%. Then, we paired the noisy version with the original modality $m_1$ to obtain a bi-modal data. For each of these data, we train CoDE-VAE assuming $\rho = 0$ and $\rho = 0.9$ and generate the non-noisy version of $m_1$. When CoDE-VAE is trained with the data with 0% noise, both modalities are the same and we expect that the model assuming $\rho = 0.9$ will have relatively high generative quality. On the other hand, when CoDE-VAE is trained with the data with 95% noise, the modalities are uncorrelated and using $\rho = 0$ is expected to have relatively high generative quality. Table 12 shows that CoDE-VAE correctly captures the dependency between experts distributions through the $\rho$ parameter.

*Table 13.* HSV color boundaries used in the coherence test of the CUB data.

|  | White | Yellow | Blue | Red | |
|---|---|---|---|---|---|
| lower bound | [0,0,120] | [25,50,70] | [90,50,70] | [0,50,70] | [159,50,70] |
| upper bound | [180,18,255] | [35,255,255] | [158,255,255] | [15,255,255] | [180,255,255] |
|  | Green | Gray | Brown | Black | |
| lower bound | [36,50,70] | [0,0,50] | [24,255,255] | [0,0,0] | |
| upper bound | [89,255,255] | [180,18,120] | [16,50,70] | [180,255,50] | |

*Table 14.* Encoder and decoder layers for captions in the CUB data. The last column for each model specifies the kernel size, stride, padding, and dilation. Layers are dense with linar activations (linear), 2D convolutional (conv), and upconvolutional (upconv) with ReLU activations. Finally, the number of input and output dimensions in each layer is shown in the columns #F.In and #F.Out, respectively.

| Encoder | | | | | Decoder | | | | |
|---|---|---|---|---|---|---|---|---|---|
| Layer | Type | #F.In | #F.Out | Spec. | Layer | Type | #F.In | #F.Out | Spec. |
| 1 | linear | 1590 | 128 | | 1 | upconv | 64 | 512 | (4,1,0,1) |
| 2 | conv | 128 | 32 | (4,2,1,1) | 2 | upconv | 512 | 256 | ((1,4),(1,2),1,1) |
| 3 | conv | 32 | 64 | (4,2,1,1) | 3 | upconv | 256 | 256 | (3,1,1,1) |
| 4 | conv | 64 | 128 | (4,2,1,1) | 4 | upconv | 256 | 128 | ((1,4),(1,2),1,1) |
| 5 | conv | 128 | 256 | ((1,4),(1,2),1,1) | 5 | upconv | 128 | 128 | (3,1,1,1) |
| 6 | conv | 256 | 512 | ((1,4),(1,2),1,1) | 5 | upconv | 128 | 64 | (4,2,1,1) |
| 7a | conv | 512 | 64 | (4,1,0,1) | 6 | upconv | 64 | 32 | (4,2,1,1) |
| 7b | conv | 512 | 64 | (4,1,0,1) | 7 | upconv | 32 | 1 | (4,2,1,1) |
| | | | | | 8 | linear | 1 | 1590 | |

*Table 15.* Comparison of generative coherence, generative quality, and classification accuracy as a function of $\rho$ on the MNIST-SVHN-Text dataset.

| | Coherence ↑ | | FID ↓ | | |
|---|---|---|---|---|---|
| $\rho$ | Unconditional | Conditional | Unconditional | Conditional | Classification ↑ |
| 0.0 | 0.52 | 0.70 | 78.2 | 95.3 | 0.88 |
| 0.2 | 0.53 | 0.78 | 77.5 | 87.4 | 0.93 |
| 0.4 | 0.51 | 0.82 | 76.7 | 83.6 | 0.95 |
| 0.6 | 0.55 | 0.81 | 79.4 | 87.0 | 0.95 |
| 0.8 | 0.54 | 0.77 | 77.0 | 91.2 | 0.95 |

