# OpenReview forum: "Aggregation of Dependent Expert Distributions in Multimodal Variational Autoencoders"
_ICML.cc/2025/Conference — ICML 2025 poster_

### Official Review · Reviewer_kHs1 · 2025-03-06

**Overall Recommendation:** 3

**Summary:**

The authors propose to challenge the assumption of independence between unimodal experts in computing the joint posterior in multimodal VAEs. Therefore they propose the CoDE-VAE that uses a Bayesian approach to compute the joint posterior between unimodal experts, modelling the dependence between them. Experimental results show that their idea is effective and are positive, despite not significantly outperforming the most recent alternative approaches.

**Claims And Evidence:**

- Empirical results are positive and show the effectiveness of the approach, despite not outperforming SOTA in the multimodal VAE literature.
- I think there seems to be some confusion in the paper about the concept of subsampling modalities in the ELBO, related to the limitations highlighted in [1]. The authors state "It is noteworthy that CODE-VAE does not rely on sub-sampling techniques, which have been shown to harm the performance of multimodal VAEs", but in the CoDE-VAE ELBO in eqn 3 subsampling actually happens. To see it, it is sufficient to notice that computing a given term of the sum in the ELBO requires reconstruction of all modalities given only a subset used for inference, hence sub-sampling of modalities happens and the CoDE-VAE is also subject to the theoretical limitations outlined in [1], as also confirmed in the experimental results.

[1] Daunhawer et al On the limitations of multimodal VAEs, ICLR, 2022.

**Essential References Not Discussed:**

Recent relevant work is not discussed. Specifically, [2]. As I mentioned above I strongly suggest the authors to at least discuss this paper in the related work, and advise also to include it in the experimental comparisons.

[2] Palumbo et al. Deep Generative Clustering with Multimodal Diffusion Variational Autoencoders, ICLR, 2024.

**Experimental Designs Or Analyses:**

The datasets chosen for the experiments are fairly standard in the multimodal VAE literature and existing models already achieve convincing results in these setups. While the comparisons on these datasets are valid, I think the authors could have picked a novel more challenging dataset to outline the benefit of their proposed model. I think experiments on the chosen datasets are properly conducted, and the results are properly commented. While model performance does not surpass certain recent approaches (e.g. MMVAE+), the results are still somehow positive and show that the suggested idea works. I'd strongly suggest the authors to compare with a recent paper [2] that proves to outperform alternative multimodal VAEs. While the proposed model has the option to be equipped with diffusion decoders, which would make an unfair comparison in terms of generative quality with CoDE-VAE, the authors show that the proposed ELBO without diffusion decoders still improves over alternative multimodal VAEs. Hence it seems relevant to include it in the comparisons in this paper.

[2] Palumbo et al. Deep Generative Clustering with Multimodal Diffusion Variational Autoencoders, ICLR, 2024.

**Methods And Evaluation Criteria:**

As outlined also below the chosen datasets to benchmark the approach are sensible, while already well-studied in the multimodal VAE literature. As for the proposed method, challenging the assumption of independence between unimodal experts in approximating joint posterior inference  for multimodal VAEs is a valuable research direction. Moreover, the proposed method appears to be effective.

**Other Comments Or Suggestions:**

- I think clarity in the paper could be improved in section 3.
- I suggest that, when grid-searching hyperparamters, the authors make explicit (e.g. in the MNIST-SVHN-Text experiment) which hyperparameters achieve best performance for each model, and hence are used to report the results.  These info can also be left to the Appendix.

**Other Strengths And Weaknesses:**

Weaknesses:
- Notation is sometimes confusing. E.g in section 3 the dimension $d$ appears at times as superscript and at times as subscript, even in the same equation ( $e^d_j = \mu^d_j - \theta_d^k$).
- Certain experimental comparisons could be more thorough. For instance, on PolyMNIST it would be more appropriate to assess the generative quality gap on all modalities (and possibly do an average in performance), instead of only focusing on generating modality $m0$.

**Questions For Authors:**

- Is the dependency between expert errors assumed to be constant across the D latent dimensions? Why do the authors make this assumption?
- Which beta values and latent space dimensions are chosen to get the results for the MMVAE+ model on the MNIST-SVHN-Text dataset, reported in the main text? It is not really clear to me, even after having a look at the Appendix. It seems somehow strange that the model achieves a relatively low performance in this dataset, keeping in mind  the results on PolyMNIST and CUB datasets.

**Relation To Broader Scientific Literature:**

The idea discussed in this paper fits nicely in the literature of multimodal VAEs as it explores the direction of modelling dependence between unimodal experts in computing the joint posterior, which was not explored thus far to my knowledge.

**Theoretical Claims:**

The authors justify their approach as a Bayesian method to approximate the joint posterior assuming a dependence between unimodal experts. The derivations to be seem correct and back up their theoretical claims.

---

> ### Author Rebuttal · Authors · 2025-04-01
>
> We appreciate your thoughtful and detailed comments. We will respond to your concerns and questions point by point.
>
> ## Confusion about the concept of sub-sampling modalities
> Thank you for pointing this out. In the paper, we use the concept of sub-sampling to refer to ELBO sub-sampling and to the use of mixture distributions to approximate consensus distributions. We will make this clear in the revised version.
>
> ## Alternative Datasets
> Based on your and Reviewer FV9Z comment, we provide results on the CELEB-A data. Due to the limited time, we only considered a latent space with 32D. For CoDE-VAE we assume $\rho=0.6$, as we observed to be a value that performs well in other experiments. We obtained the following results
>
> |                           | CoDE-VAE  | MMVAE+    |
> |---------------------------|-----------|-----------|
> | Conditional FID           | 92.11 (0.61)      |  97.30 (0.40)    |
> | Unconditional FID         | 87.41 (0.36)      |  96.91 (0.42)    |
> | Conditional Coherence     | 0.38 (0.001)      |  0.46 (0.001)    |
> | Unconditional Coherence   | 0.23 (0.003)      |  0.31 (0.030)    |
> | Classification            | 0.38 (0.066)      |  0.37 (0.003)    |
>
> which may improve further by cross-validating $\rho$ in CoDE-VAE.
>
> ## Comparison to Deep Generative Clustering with Multimodal Diffusion Variational Autoencoders
> Thank you for pointing out this paper, which we were not aware of. We acknowledge the importance of the paper in the field of multimodal VAEs and will discuss it in the related work section and will include the experimental comparison on PolyMNIST in the appendix of the revised version of the paper, as the Clustering Multimodal VAE (CMVAE) model introduced in [2] is not 100\% comparable with our proposed CoDE-VAE model. The main focus of our research is to present a novel approach to estimate consensus distributions and to learn the contribution of each ELBO subset, while the goal in CMVAE is to couple multimodal VAEs with clustering tasks by leveraging clustering structures in the latent space and to introduce diffusion decoders, which is certainly a novel and relevant line of research. CMVAE captures clustering structures using a mixture model as a prior, and we hypothesized that this flexible prior in CMVAE plays an important role in the performance of unconditional generative tasks. We will include this discussion and the relation between the methods in the updated manuscript.
>
> ## Notation and Clarity
> Thank you for pointing this out. We will change the notation in the revised version to ensure consistency in the use of subscripts and superscripts, which will improve the clarity of Section 3.
>
> ## Thorough comparison on the generative quality gap for PolyMNIST
> We agree that an average generative quality gap would provide a robust comparison. However, the computational costs of such experiment is significant, as to assess the generative quality for each of the 5 modalities as a function of the number of input modalities requires to train at least 12*3=36 times each model (considering 3 different runs to report standard deviations). This will require 252 runs in total, as there are 7 different models in the evaluation of the quality gap (without considering the unimodal VAEs). Given that our research includes 3 data sets, 6 benchmark models, and several ablation experiments, we let such a robust comparison for future research.
>
> ## Add grid-search hyperparameters
> Thank you for pointing this out. We will add to the Appendix the hyperparameters for the grid search and their optimal values, including the ones for the new experiments on the CELEB-A data.
>
> ## Is the dependency between expert errors assumed to be constant across the D latent dimensions?
> Yes, for simplicity, we assume a common $\rho$ parameter for all dimensions. However, the CoDE is a flexible approach that does not impose any restriction on the way $\Sigma_d$ is specified. We let future research explore whether model performance can be improved by using different correlation values for different dimension.
>
> ## Beta values and latent space for MMVAE+ on the MNIST-SVHN-Text data.
> For all models in Section 4.1, we cross-validate $\beta$ using the grid $[0.1,1,5,10,15,20]$. All models, but MMVAE+, assume that the latent space has 20 dimensions as in previous research. To select the dimensionalty of common and modality specific (MS) variables in MMVAE+, we follow a similar approach as in the original paper, where the authors choose the dimensions of these two to be equal to the dimension of the latent space in MMVAE (a model without MS variables) divided by the number of modalities. Therefore, MMVAE+ assumes that both common and MS variables have 7 dimensions. These details are explained in Appendix D3. We also tested to use 10 dimensions in both common and MS variables, so the decoders in the MMVAE+ model would generate modalities based on 20 dimensions, just as the other models. We did not observe significant differences.

---

### Official Review · Reviewer_kGwL · 2025-03-11

**Overall Recommendation:** 3

**Summary:**

The paper introduces a new method for aggregating multimodal expert distributions in Variational Autoencoders (VAEs) by incorporating the dependence between experts, which has traditionally been ignored in models like the product of experts (PoE) and mixture of experts (MoE).  This method, called Consensus of Dependent Experts (CoDE), aims to improve the estimation of joint likelihoods in multimodal data by accounting for the dependencies between different modality-specific distributions.  The paper proposes the CoDE-VAE model, which enhances the trade-off between generative coherence and quality, improving log-likelihood estimations and classification accuracy.  The authors claim that CoDE-VAE performs better than existing multimodal VAEs, especially as the number of modalities increases.

**Claims And Evidence:**

The paper provides clear empirical evidence to support the claims about the CoDE-VAE’s superior performance.  The experimental results on datasets such as MNIST-SVHN-Text, PolyMNIST, and CUB support the assertion that CoDE-VAE balances generative coherence and quality better than existing models.  Additionally, the paper argues that CoDE’s consideration of expert dependence leads to better log-likelihood estimations and reduced generative quality gaps compared to models relying on modality sub-sampling.  However, the discussion could benefit from more in-depth comparisons in specific edge cases where other models might outperform CoDE-VAE.

**Essential References Not Discussed:**

No.

**Experimental Designs Or Analyses:**

The experimental design is robust, with comprehensive comparisons to multiple baseline models.  The use of multiple datasets (MNIST-SVHN-Text, PolyMNIST, and CUB) provides a good cross-section of real-world multimodal problems.  However, more detailed ablation studies or analyses of edge cases where the assumptions about expert dependence may not hold could further strengthen the paper.

**Methods And Evaluation Criteria:**

The methodology behind CoDE is sound, introducing a principled Bayesian approach to account for expert dependence.  The CoDE-VAE model builds on existing multimodal VAEs, addressing key challenges like missing modalities and the imbalance in the contribution of different ELBO terms.  The evaluation criteria, including generative coherence, log-likelihood estimation, and classification accuracy, are appropriate for comparing multimodal models.  However, the explanation of how the model behaves in extreme cases (e.g., when only one modality is available) is not fully addressed.

**Other Comments Or Suggestions:**

1. Consider adding more analysis on how CoDE-VAE behaves in cases with missing data or when some modalities are not available.
2. A clearer distinction between the CoDE-VAE approach and similar models would benefit readers unfamiliar with multimodal VAEs.

**Other Strengths And Weaknesses:**

Strengths:
1. The CoDE-VAE method is a novel and theoretically sound approach that addresses the challenge of dependent expert distributions.
2. The experimental results are convincing, showing that CoDE-VAE outperforms existing methods in key areas like generative coherence and log-likelihood estimation.

Weaknesses:
1. Some parts of the experimental setup could be explained more clearly, particularly regarding the optimization process for learning the contribution of each ELBO term.
2. More detailed comparisons with edge cases or failures of the model would help to solidify the generalizability of the results.

**Questions For Authors:**

1. How does the performance of CoDE-VAE change when there is a significant imbalance between the available modalities (e.g., when one modality is much more informative than the others)?
2. Could you elaborate on how the CoDE method handles scenarios where the assumption of expert dependence does not hold (e.g., in highly independent modalities)?
3. The paper mentions that CoDE-VAE reaches generative quality similar to unimodal VAEs in certain cases—could you provide more concrete examples of these cases, and how the model behaves with increasing modality count?
4. What are the computational complexities of CoDE-VAE compared to existing models like PoE and MoE, particularly as the number of modalities increases?

**Relation To Broader Scientific Literature:**

The paper does a good job of relating its contributions to the broader literature on multimodal VAEs and expert aggregation methods.  The work is clearly motivated by existing challenges in multimodal learning, such as missing modalities and independent expert assumptions.  The comparison with PoE and MoE methods, along with references to key multimodal VAE papers, establishes the novelty of the approach.

**Theoretical Claims:**

The theoretical claims regarding the new ELBO formulation and the aggregation method using CoDE are well-supported by the paper’s derivations and lemmas.  The paper provides a solid mathematical foundation for the method, with proofs of key results, such as the posterior distribution and consensus distributions.  There are no apparent issues with the correctness of these proofs.

---

> ### Author Rebuttal · Authors · 2025-04-01
>
> Thank you for your thoughtful and detailed comments. We will address your concerns and questions point by point.
>
> ## Experiments and analysis on edge cases
> We agree that analyzing CoDE-VAE on edge cases helps to understand its behavior, and makes our research more robust. We believe that the experiments in Section 4.2 Generative quality gap (Figure 3), in Appendix D.4. PolyMNIST (Figure 13), and the classification results in Figures 8 and 10 provide a good indication that the performance of CoDE-VAE improves with the number of modalities and with the cardinality of the subset on which consensus distributions are conditioned on.
>
> To address the scenario where the assumption about dependent experts may not hold, we trained the CoDE-VAE model on PolyMNIST using the modality $m_1$ in the following way. We apply 3 different levels of noise to the modality $m_1$, 0 \%, 25\%, and 95\%. Then, we paired the noisy version with the original modality to obtain a bi-modal data. For each of these data, we train CoDE-VAE assuming $\rho=0$ and $\rho=0.9$, and generate the non-noisy version of $m_1$. When CoDE-VAE is trained with the data with 0\% noise, both modalities are the same and we expect that $\rho=0.9$ will have relatively high generative quality. On the other hand, when CoDE-VAE is trained with the data with 95\% noise, the modalities are uncorrelated and $\rho=0$ is expected to have relatively high generative quality. We obtain the following average FID scores ([link1](https://anonymous.4open.science/r/codevae_icml-27EA/), [backup_link](https://anonymfile.com/k0W28/uncorrelated-experts.pdf)) :
>
> |               | 0\%       | 25\%      | 95\%      |
> |---------------|-----------|-----------|-----------|
> | $\rho=0$      | 29.0      | 31.27     | 48.22     |
> | $\rho=0.9$    | 26.12     | 29.90     | 53.68     |
>
> showing that CoDE-VAE correctly captures the dependency between experts distributions through the $\rho$ parameter.
>
> ## CoDE-VAE in cases with missing data or when modalities are not available.
> The evaluation setup in our research follows the standard method in multimodal VAEs where all possible combinations of missing modalities are evaluated at test time (Appendix B). To handle missing data is not trivial, as we need to estimate consensus distributions. To estimate $q(z|x_1,x_2)$, any aggregation method would require the same number of observations $x_1$ and $x_2$. This problem could be overcome by using only complete pairs $(x_1,x_2)$, and re-weighting the ELBO terms with fewer samples and could be an interesting direction to pursue in future work.
>
> ## CoDE-VAE when one modality is much more informative?
> CoDE-VAE learns the contribution of each k-th ELBO term to the optimization, balancing the importance of relatively more informative modalities. The empirical results of Section 4.4 show that the text modality in the MNIST-SVHN-Text is relatively more important to the optimization of the ELBO as shown by the weight learned for the subset containing that modality. This result seems reasonable, as there is more noise in the MNIST and SVHN modalities. The ablation experiments of Section 4.5 show that CoDE-VAE achieves higher performance (coherence and FID), when the contribution of each subsets to the optimization, and so each modality, is learned.
>
> ## CoDE where the assumption of expert dependence does not hold?
> CoDE is a flexible approach that does not impose any restriction on the way $\Sigma_d$ is specified as long as it is invertible. For independent modalities, it should be enough to use $\rho=0$ (see answer on edge cases).
>
> ## CoDE-VAE generative quality. Could you provide more concrete examples?
> We recognize that the wording of this claim could be improved and made it more concrete. We have replaced the original sentence with *"CoDE-VAE minimizes the generative quality gap as the number of modalities increases, achieving quality similar to unimodal VAEs measured by unconditional FID scores."*, which corresponds to the experiments in Section 4.2 Generative quality gap. Furthermore, Figure 3 shows that CoDE-VAE achieves higher generative performance as the number of modalities used for model training increases, something most of the benchmark models are not able to achieve. We have [added figures](https://anonymous.4open.science/r/codevae_icml-27EA/) ([backup](https://anonymfile.com/rN124/polymnist-gen.pdf)) that show the generated modality $m_0$  as a function of input modalities, as well as generated samples by the unimodal VAE for qualitative comparisons.
>
> ## Computational complexities of CoDE-VAE
> We agree that this is an important aspect to be considered. Therefore, in Appendix C we mention that CoDE-VAE has a relatively high computational cost $\mathcal{O}(2^M-1)$, where $M$ is the number of modalities. However, given that the size of the matrix $\Sigma_d$ depends only on $M$ (see the question from reviewer FV9z), model training is feasible on a single GPU even for 5-modality datasets.

---

> > ### Comment · Reviewer_kGwL · 2025-04-05
> >
> > The authors' rebuttal is quite professional and addresses some of my concerns successfully. I will be thinking about editing my initial review and rating after carefully going through other rebuttal contents to other reviewers (but will not require any additional details or raise questions from/to authors).

---

> > > ### Author Response · Authors · 2025-04-05
> > >
> > > Thank you for your positive feedback. We are pleased to have been able to address your concerns.

---

### Official Review · Reviewer_FV9z · 2025-03-14

**Overall Recommendation:** 4

**Summary:**

This paper introduces the Consensus of Dependent Experts (CoDE) in the context of multimodal learning with Variational Autoencoders (VAEs). Current approaches for this task, such as: (i) the product of experts; or (ii) the mixture of experts assume cross-modal independence which is restrictive. Towards this end, the current work proposes a novel Empirical Lower Bound (ELBO) that estimates the joint likelihood by learning the contribution of each modality. The proposed method can strike a balance between generative coherence and generative quality. Empirical evaluations are conducted on several datasets.

## Update after rebuttal
Convinced with the responses to my questions by the authors. Hence, raising my score.

**Claims And Evidence:**

Generally, the claims make sense. However, the chief concern with the claims is:
1. The paper does not show how accurately the ELBO is minimized for the different datasets. This is an important shortcoming of the present work.

**Essential References Not Discussed:**

Most important related works have been discussed.

**Experimental Designs Or Analyses:**

Overall, the experimental evaluation is pretty broad but has the following shortcomings:
1. Some of the more complex real-world image datasets, CELEB-A [a], CELEB-HQ [b] have not been experimented with.
2. Generation Quality (as measured by FID scores) and Classification accuracy are not the best.

References:
[a] Liu, Z., Luo, P., Wang, X. and Tang, X., 2018. Large-scale celebfaces attributes (celeba) dataset. Retrieved August, 15(2018), p.11.
[b] Karras, T., Aila, T., Laine, S. and Lehtinen, J., 2018, February. Progressive Growing of GANs for Improved Quality, Stability, and Variation. In International Conference on Learning Representations.

**Methods And Evaluation Criteria:**

While overall the method is intuitive, the chief concern is as follows:
1. L201, Col 2: Estimating the off-diagonal elements in $\Sigma_d$ in the forward pass could be computationally expensive for high-dimensional scenarios.
2. Also, how does the model deal with cases, where $\Sigma_d$ is not full rank.

**Other Comments Or Suggestions:**

The following are some additional comments:
1. L98, Col 1: "....method, the derivation..." -> "....method, for the derivation..."
2. L376, Col 1: "...significant..." -> "...significantly..."

**Other Strengths And Weaknesses:**

Other strengths:
1. The proposed model can deal with scenarios when one or more modalities are missing.
2. The current formulation allows for the estimation of uncertainties of each of the experts.

Other weaknesses:
See previous sections.

**Questions For Authors:**

The following are some questions for the authors:
1. L89, Col 1: Does the proposed approach really "minimize the generated quality as the number of modalities increase". This seems counter-intuitive. Or is this a typo?
2. What would happen if instead of a Categorical distribution, a softmax distribution is used to weigh the experts?

**Relation To Broader Scientific Literature:**

The idea of combining multimodal distributions relate to prior works that combine hidden markov model distributions (Brown & Hinton, 2001), image synthesis to combine modalities and generate images using generative adversarial networks (Huang et al., 2022), large language models for comparative assessment of texts (Liusie et al., 2024), in early-exit ensembles (Allingham & Nalisnick, 2022), or in diffusion models to aggregate distillation of diffusion policies (Zhou et al., 2024).

**Theoretical Claims:**

Generally, the theoretical claims seem accurate.

---

> ### Author Rebuttal · Authors · 2025-04-01
>
> Thank you for your careful and comprehensive comments. We will address your concerns and questions, point by point:
>
> ## How accurately is ELBO minimized:
> We are not completely sure if we follow your concern. If the comment refers to how the ELBO is maximized (the loss), we train the CoDE-VAE model until the ELBO converges, confirmed by visual inspection. [These plots](https://anonymous.4open.science/r/codevae_icml-27EA/) ([backup](https://anonymfile.com/LNara/elbo-cub.png) [backup](https://anonymfile.com/Vxpdx/elbo-mmnist.png) [backup](https://anonymfile.com/8p9jO/elbo-mst.png)) show the convergence of the ELBO. If your concern is about how close the ELBO is to the intractable marginal log-likelihood, we calculate log-likelihoods on the test sets using importance-sampling (shown in Figures 2 and 4 for all models). Please let us know If we are misunderstanding your concern.
>
> ## $\Sigma_d^{-1}$ costly in high-dimensional data.
> $\Sigma_d$ is not a sample covariance matrix and its size depends on the number of expert distributions assessing consensus distributions (CD). So, it is limited by the number of modalities M, which typically is small and only one CD is conditioned on all modalities. In our research the largest M is 5 (PolyMNIST). Therefore, the computational costs to find the inverse of $\Sigma_d$ are affordable.
>
> ## $\Sigma_d$ if it is not full rank.
> $\Sigma_d$ is guaranteed to be full rank by construction, as $\sigma_{i,j}>0$ for all $i,j$, where $\sigma_{i,i}=\sigma^2_i$. To see this, we need to show that the quadratic form $\beta^T\Sigma_d\beta = 0$, is only satisfied for a zero-vector $\beta$. Let $\kappa$ be the smallest $\sigma_{i,j}$ value, which is positive by construction. Therefore, $\sum_i \sum_j \beta_i \sigma_{i,j} \beta_j > \kappa \sum_i \sum_j \beta_i \beta_j$. Since $\kappa>0$, the only solution that satisfies $\kappa \sum_i \sum_j \beta_i \beta_j=0$ is the zero-vector $\beta$. We will add this discussion in the revised version.
>
> ## More complex datasets
> We added a new section in the appendix with experiments on CELEB-A for CoDE-VAE and MMVAE+, which is the model that stands out in the other experiments. Due to the limited time, we evaluate both models only for $\beta=1$, assuming a latent space with 32D. For CoDE-VAE we assume $\rho=0.6$, as we observed in other experiments to be a value that performs consistently well. We obtained the following results:
>
> |                           | CoDE-VAE  | MMVAE+    |
> |---------------------------|-----------|-----------|
> | Conditional FID           | 92.11 (0.61)      |  97.30 (0.40)    |
> | Unconditional FID         | 87.41 (0.36)      |  96.91 (0.42)    |
> | Conditional Coherence     | 0.38 (0.001)      |  0.46 (0.001)    |
> | Unconditional Coherence   | 0.23 (0.003)      |  0.31 (0.030)    |
> | Classification            | 0.38 (0.066)      |  0.37 (0.003)    |
>
> which may improve further by cross-validating $\rho$ in CoDE-VAE.
>
> ## Generation Quality and Classification accuracy
> Multimodal VAEs trade high generative quality with reduced generative coherence [1]. The experimental setup in our research shows that CoDE-VAE performs as well as or better than SOTA multimodal VAEs in terms of balancing the trade off between generative coherence and generative quality. When we assess in isolation whether multimodal VAEs are able to improve generative quality as the number of modalities increases, CoDE-VAE clearly shows a higher performance (Figure 3). It is possible to add modality-specific (MS) latent variables to CoDE-VAE to further improve the generative quality, which requires a careful design of the number of dimensions in the common and MS variables to avoid the short-cut problem [1]. When it comes to the classification results, Figures 8 and 10 in the appendix show that models using the mixture-of-experts are not able to achieve higher classification accuracy as latent representations are learned from subsets with more modalities. On the other hand, CoDE-VAE ranks 1st and 3rd, while balancing the trade-off between generative quality and generative coherence at the same time.
>
> [1] Palumbo et al. Enhancing the Generative Quality of Multimodal VAEs Without Compromises, ICLR, 2023.
>
> ## Typos and L89, Col 1
> You are correct that there is a typo in L89, Col 1. The sentence should read *"Furthermore, CoDE-VAE minimizes the generative quality gap as the number of modalities increases..."*, referring to the results in Figure 3. We will fix this in our revised version of the paper.
>
> ## Softmax instead of categorical
> We assume that you are referring to the Gumbel-Softmax distribution, which is useful when we need to sample and backpropagate at the same time. As our main interest is learning the $\pi$ parameters, we do not expect a significant different behavior by using the Gumbel-Softmax distribution. The main concern when using a different distribution, is whether the entropy term in the ELBO of the CoDE-VAE model can be evaluated, ideally in closed form.

---

> > ### Comment · Reviewer_FV9z · 2025-04-07
> >
> > I am grateful to the authors for addressing my concerns.

---

> > > ### Author Response · Authors · 2025-04-08
> > >
> > > Thank you for your positive feedback. We're glad we were able to address your concerns and hope you'll consider reviewing your score based on our rebuttal.

---

### Decision · Program_Chairs · 2025-05-01

**Decision:**

Accept (poster)

**Comment:**

This work  introduces CoDE-VAE, a model that challenges the common assumption of independence between unimodal experts in multimodal Variational Autoencoders by modeling their dependencies. Using a Bayesian approach, CoDE-VAE aims to improve joint posterior estimation and enhance generative coherence and classification accuracy. Experimental results show that while CoDE-VAE performs well, its improvements over the latest alternatives are modest.

While this work has some limitations, like including more thorough experiments, improving clarity of the paper overall, and more detailed comparisons with edge cases or failures of the model would help to solidify the generalizability of the results, the strength outweigh the limitations. The authors propose a method that is novel and theoretically sound, and it addresses the challenge of dependent expert distributions.  I think this work is of interest to the ML community